# $Pt_n$–$O_v$ synergistic sites on $MoO_x$/$\gamma$-$Mo_2N$ heterostructure for low-temperature reverse water–gas shift reaction

Hao-Xin Liu [1,6], Jin-Ying Li [1,6], Xuetao Qin [2,6], Chao Ma [3], Wei-Wei Wang [1], Kai Xu [1], Han Yan [1], Dequan Xiao [4], Chun-Jiang Jia [1] ✉, Qiang Fu [5] ✉ & Ding Ma [2] ✉

In heterogeneous catalysis, the interface between active metal and support plays a key role in catalyzing various reactions. Specially, the synergistic effect between active metals and oxygen vacancies on support can greatly promote catalytic efficiency. However, the construction of high-density metal-vacancy synergistic sites on catalyst surface is very challenging. In this work, isolated Pt atoms are first deposited onto a very thin-layer of $MoO_3$ surface stabilized on $\gamma$-$Mo_2N$. Subsequently, the Pt–$MoO_x$/$\gamma$-$Mo_2N$ catalyst, containing abundant Pt cluster-oxygen vacancy ($Pt_n$–$O_v$) sites, is in situ constructed. This catalyst exhibits an unmatched activity and excellent stability in the reverse water-gas shift (RWGS) reaction at low temperature (300 °C). Systematic in situ characterizations illustrate that the $MoO_3$ structure on the $\gamma$-$Mo_2N$ surface can be easily reduced into $MoO_x$ ($2 < x < 3$), followed by the creation of sufficient oxygen vacancies. The Pt atoms are bonded with oxygen atoms of $MoO_x$, and stable Pt clusters are formed. These high-density $Pt_n$–$O_v$ active sites greatly promote the catalytic activity. This strategy of constructing metal-vacancy synergistic sites provides valuable insights for developing efficient supported catalysts.

The design of efficient supported catalysts has been a frontier research in catalysis. For heterogeneous catalytic reactions, due to the complexity and diversity of reactant molecules, high-performance supported catalysts always require the cooperation of multiple components[1–4]. At the metal-support interface, the active metal and the support work collectively to activate different reactant molecules to form active intermediates, which can greatly facilitate a variety of important catalytic reactions, such as CO oxidation[5], water–gas shift reaction[6], and $CO_2$ reduction[7]. Therefore, the construction of active interfaces is essential for the design of efficient supported catalysts.

However, the catalytic properties of interfacial structure can be influenced by many factors, such as the size of active metals[1,8], the type of supports[9], the strength of metal-support interactions[10], making targeted creation of active interfaces with specific structure a challenge.

For reducible supports, the oxygen vacancy on surfaces has attracted extensive attention in the field of catalysis[11–16]. Importantly, interfacial oxygen vacancies can form special active sites in cooperation with neighboring active metals, promoting catalytic reactions effectively. Recently, the synergistic effect of active metals and vacancies has been demonstrated[5,6,17]. For active metal-oxygen vacancy

[1]Key Laboratory for Colloid and Interface Chemistry, Key Laboratory of Special Aggregated Materials, School of Chemistry and Chemical Engineering, Shandong University, Jinan 250100, China. [2]College of Chemistry and Molecular Engineering, Peking University, Beijing 100871, China. [3]College of Materials Science and Engineering, Hunan University, Changsha 410082, China. [4]Center for Integrative Materials Discovery, Department of Chemistry and Chemical and Biomedical Engineering, University of New Haven, West Haven, CT 06516, USA. [5]School of Future Technology, University of Science and Technology of China, Hefei 230026, China. [6]These authors contributed equally: Hao-Xin Liu, Jin-Ying Li, Xuetao Qin. ✉e-mail: jiacj@sdu.edu.cn; qfu3@ustc.edu.cn; dma@pku.edu.cn

synergistic site, the oxygen vacancy plays a crucial role in anchoring and activating the active metal, thus stabilizing interfacial structures and accelerating the activation of reactant molecules. So far, research works on the synergistic effect between active metal and oxygen vacancy have been mostly focused on the catalysts using reducible oxides as support. However, due to the stability of the surface oxygen atoms, the ability to generate surface oxygen vacancies in these oxides has been restricted[18–20], which undoubtedly limits the creation of active metal-oxygen vacancy synergistic sites. Besides oxide supports, transition-metal nitrides like $Mo_2N$ have received extensive attention in a variety of catalytic reactions due to their unique electronic structure[9]. Due to the contraction of the $d$-band caused by the interstitial incorporation of N in the Mo metal lattice, $Mo_2N$ exhibited a noble-metal-like electronic structure. In order to prevent the spontaneous combustion of molybdenum nitrides in air, it is often necessary to slowly oxidize its surface during the preparation process. Under the effect of surface heterostructural stress, the molybdenum oxide surface was easily reduced[21], providing the potential to solve the problem of limited oxygen vacancy concentration on the surfaces of conventional oxides.

The reverse water–gas shift (RWGS) reaction is an important reaction to utilize $CO_2$[22–25]. The product CO can be further converted into high-value chemicals through Fischer–Tropsch synthesis[26–28]. However, $CO_2$ molecule is very stable, and the RWGS reaction is endothermic, making the conversion of $CO_2$ at low temperature very challenging. In general, for the RWGS catalysts, active metal clusters can dissociate $H_2$, while the oxygen vacancies can activate $CO_2$. Therefore, the creation of catalysts with high-density synergistic sites between active metal cluster and oxygen vacancy may greatly improve the low-temperature activity of catalysts. Herein, we deposited single Pt atoms on the $\gamma$-$Mo_2N$ support with a $MoO_3$ passivation surface. Based on the stress of $MoO_3/\gamma$-$Mo_2N$ heterostructure, the surface $MoO_3$ was transformed into $MoO_x$ structure with the creation of sufficient surface oxygen vacancies in the pretreatment and reaction processes. Meanwhile, isolated Pt atoms were reduced to form Pt clusters with the size less than 1 nm, which were anchored on the $MoO_x$ surface to form abundant and stable metal cluster-oxygen vacancy synergistic sites. The obtained Pt–$MoO_x/\gamma$-$Mo_2N$ catalyst exhibited unmatched catalytic performance with a CO yield of $17.2 \times 10^{-5}$ mol $g_{cat}^{-1}$ $s^{-1}$ at 300 °C, to catalyze the low-temperature reverse water–gas shift (RWGS) reaction, which far exceeded those of previously reported catalysts. Comprehensive in situ characterizations and theoretical calculations revealed that the $Pt_n$–$O_v$ synergistic sites enhanced the activation of reactant molecules. This strategy of constructing high-density active metal-vacancy interfacial sites using the stress of heterostructure and metal–support interactions provides a new way to enhance the activity of supported catalysts.

## Results

### Structure characterization of the Pt–$MoO_3/\gamma$-$Mo_2N$ catalyst

The molybdenum oxide structure on the $\gamma$-$Mo_2N$ surface was the safeguard against spontaneous combustion of $\gamma$-$Mo_2N$ in air. During the preparation of $\gamma$-$Mo_2N$ sheet, the surface of $\gamma$-$Mo_2N$ was slightly oxidized by passivation treatment with low concentration of oxygen (1% $O_2$/Ar) at room temperature to form the special $MoO_3/\gamma$-$Mo_2N$ heterostructure. The stress of $MoO_3/\gamma$-$Mo_2N$ heterostructure made $MoO_3$ highly susceptible to deoxygenation to form $MoO_x$ ($2 < x < 3$) surface with rich oxygen vacancies[21]. After the loading of Pt, the $MoO_x$ surface provided the possibility for the construction of high density Pt-oxygen vacancy sites. In order to verify that the as-prepared catalyst had the expected structure as shown in Fig. 1a, comprehensive characterizations were carried out. The transmission electron micrograph (TEM) images exhibited that all fresh and used catalysts had sheet structures (Supplementary Fig. 1). To explore the morphology and particle size of catalysts, high-angle annular dark-field scanning TEM

measurements (HAADF-STEM) were conducted. For fresh 0.5Pt–$MoO_3$–$Mo_2N$, the porous $\gamma$-$Mo_2N$ nanosheet was composed of numerous $Mo_2N$ particles with diameters ranging from 3 to 5 nm, and isolated Pt atoms were anchored on the $\gamma$-$Mo_2N$ support (Fig. 1b and Supplementary Fig. 2). As illustrated in Fig. 1c and Supplementary Fig. 3, most of the Pt atoms with Pt loading of 0.59 wt.% (Supplementary Table 1) were transformed to Pt clusters with diameters less than 1 nm after the RWGS reaction, suggesting the aggregation of Pt atoms during the reduction pretreatment. The element mapping results indicated the weak surface oxidation of $Mo_2N$, with O signal appearing neared the Mo and N elements (Supplementary Fig. 4). The HAADF-STEM results of fresh and used 0.5Pt–$MoO_3/\gamma$-$Mo_2N$ catalysts indicated that the bulk structure of $\gamma$-$Mo_2N$ support was stable during the reaction. As shown in Supplementary Figs. 5a and 6, for both the fresh and used catalysts, only X-ray diffraction (XRD) patterns of $\gamma$-$Mo_2N$ were detected without any discernible peak of Pt species, implying that Pt species were highly dispersed on the support. However, different from the XRD results, Raman spectra of the samples before and after the RWGS reaction only showed the signal of $MoO_3$[21]. Thus, the surface of the catalyst was covered with a thin layer of $MoO_3$ passivation structure in the air (Supplementary Figs. 5b and 7), which was consistent with the element mapping result. Furthermore, the in situ Raman study of 0.5Pt–$MoO_3/\gamma$-$Mo_2N$ illustrated that the surface $MoO_3$ was transformed to $MoO_x$ by deoxygenation in the RWGS reaction (Supplementary Fig. 5b), accompanied by the formation of abundant oxygen vacancies.

In order to explore the local structure of Pt–$MoO_3/\gamma$-$Mo_2N$, X-ray absorption fine structure (XAFS) spectra were measured. As shown in Fig. 1d, the extended X-ray absorption fine structure (EXAFS) spectra showed only one peak at ~1.7 Å as the Pt-O contribution in fresh 0.5Pt–$MoO_3/\gamma$-$Mo_2N$, while the peak at ~2.6 Å from Pt–Pt contribution was dominant in the used sample. Thus, the single Pt atoms in 0.5Pt-$MoO_3/\gamma$-$Mo_2N$ agglomerated into nanoclusters during the RWGS reaction, which was consistent with the HAADF-STEM results. For the Pt-$MoO_3/\gamma$-$Mo_2N$ catalysts with higher Pt loadings (1 wt. % and 2 wt. %) before and after the reaction, in addition to the Pt–O coordination, Pt–Pt bond was observed, suggesting the presence of Pt clusters (Supplementary Fig. 8). After the reaction, the Pt–Pt contribution increased, indicating the reduction of Pt single atoms (Supplementary Figs. 8 and 9). Meanwhile, the small coordination numbers of Pt–Pt bond (≤3.0) in all samples suggested that Pt species were highly dispersed on the support without obvious aggregation into nanoparticles (Supplementary Fig. 10 and Supplementary Table 2). The presence of Pt–O peak indicated that Pt was bound with O atoms rather than Mo atoms. Supplementary Fig. 11 showed the X-ray absorption near edge spectra (XANES) of fresh Pt–$MoO_3/\gamma$-$Mo_2N$ catalysts. The white line intensities in the spectra of all catalysts were between that of $PtO_2$ and Pt foil, suggesting that Pt species were positively charged. With the increasing of Pt loading, more metallic Pt was detected. The XANES spectra of Pt–$MoO_3/\gamma$-$Mo_2N$ (Fig. 1e) showed that the oxidized Pt species were partially reduced during the RWGS reaction, which is consistent with the EXAFS results. Based on the above results, we proposed the following structure of catalyst (Fig. 1a): a thin layer of $MoO_x$ with abundant oxygen vacancies was in situ formed from $MoO_3$ on the surface of the $\gamma$-$Mo_2N$, with Pt clusters anchored on $MoO_x$. The $MoO_x$ with abundant oxygen vacancies provided a platform for the construction of sufficient platinum cluster-oxygen vacancy synergistic sites.

### Catalytic performance of the Pt–$MoO_x/\gamma$-$Mo_2N$ catalysts in the RWGS reaction

The catalytic performances of the Pt–$MoO_x/\gamma$-$Mo_2N$ catalysts for the RWGS reaction were evaluated at various temperatures under a very high space velocity of 300,000 mL $g_{cat}^{-1}$ $h^{-1}$. As illustrated in Fig. 2a, the pure $\gamma$-$Mo_2N$ support was fully inactive at relatively low temperatures

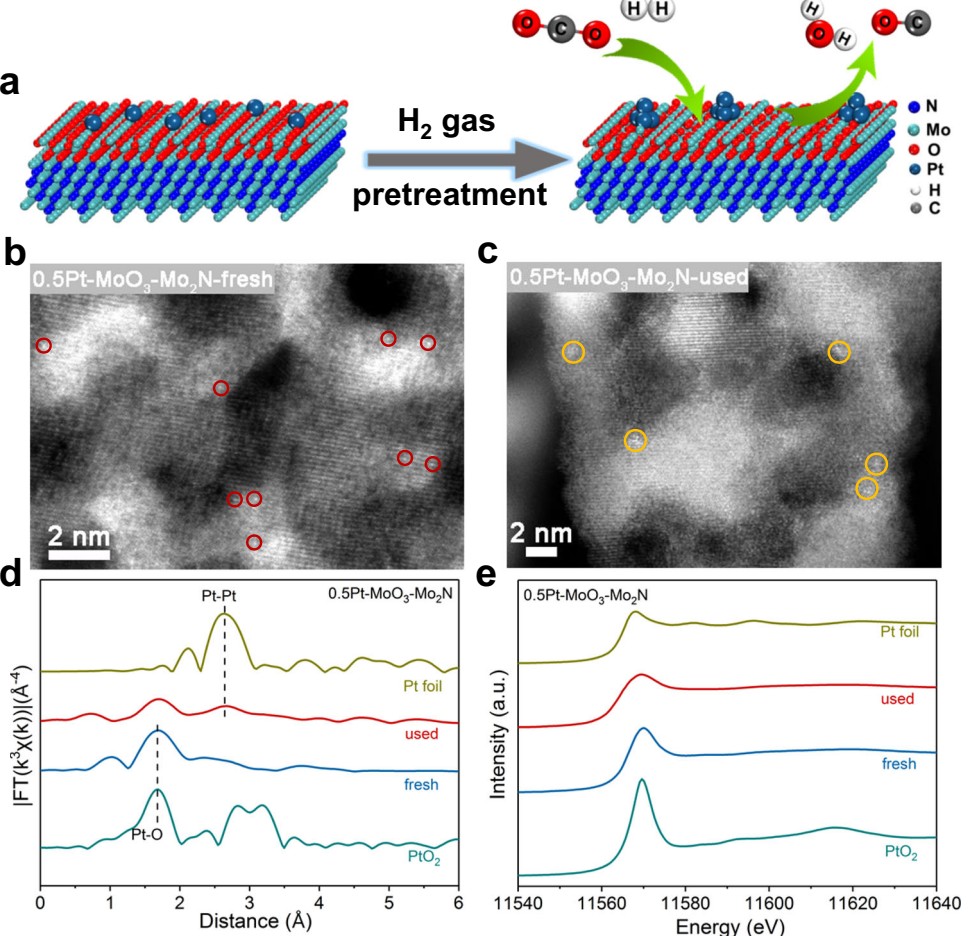

**Fig. 1 | Structural characterization of 0.5Pt–MoO₃/Mo₂N. a** Schematic illustration of the catalyst structure. **b**, **c** Aberration-corrected HAADF-STEM images of fresh and used catalysts. **d**, **e** EXAFS and XANES spectra of Pt L₃ edge for fresh and used catalysts, respectively.

(<300 °C), indicating that the $MoO_3/\gamma$-$Mo_2N$ support without platinum clusters could not catalyze this reaction. The deposition of Pt on $MoO_3/\gamma$-$Mo_2N$ effectively promoted the activity of catalysts at low temperatures (Fig. 2a and Supplementary Fig. 12). With the increasing of Pt loading, the catalytic efficiency increased gradually. $CO_2$ could even be converted at 200 °C when the Pt loading reached 0.5 wt.%. When the Pt loading exceeded 0.5 wt.%, the $CO_2$ conversion did not increase much. At 300 °C, the $CO_2$ conversion rate of 0.5Pt–$MoO_x$/$Mo_2N$ was close to the upper limit of thermodynamic equilibrium, which was much higher than that of the reference 2Pt–$CeO_2$ catalyst with 2 wt.% Pt loading (Fig. 2a). Importantly, the reaction rate of 0.5Pt–$MoO_x/\gamma$-$Mo_2N$ reached as high as $17.2 \times 10^{-5}$ $mol_{CO2}$ $g_{cat}^{-1}$ $s^{-1}$ at 300 °C, which was definitely the highest value compared with all other reported catalysts for low-temperature RWGS reactions. This value even exceeded those of many catalysts working under much higher temperature (350–500 °C) (Fig. 2b and Supplementary Table 3). Obviously, the Pt–$MoO_x/\gamma$-$Mo_2N$ catalyst improved the catalytic efficiency for low-temperature RWGS reactions to a new level. Furthermore, 0.5Pt–$MoO_x/\gamma$-$Mo_2N$ showed good stability, and it could maintain ~80% initial $CO_2$ conversion after 300 h reaction at 300 °C with a space velocity of 300,000 mL $g_{cat}^{-1}$ $h^{-1}$ (Fig. 2e). Besides, the apparent activation energy ($E_a$) for all Pt–$MoO_x$/$Mo_2N$ catalysts was around 40 kJ mol⁻¹ (Supplementary Fig. 13), which was only half than that of 2Pt–$CeO_2$ (~82 kJ mol⁻¹), indicating the great catalytic efficiency of Pt–$MoO_x$/$Mo_2N$ (Fig. 2c). As shown in Fig. 2d, the reaction orders of $H_2$ were 0.43, 0.55 and 0.61 at 250, 280, and 300 °C, respectively. The reaction orders of $CO_2$ were 0.07, 0.20, and 0.26 at 250, 280, and

300 °C, respectively. The reaction orders of $CO_2$ and $H_2$ increased with rising temperature, suggesting that the adsorption of reactant molecules became difficult with rising temperature. Meanwhile, the low $CO_2$ reaction order meant that $CO_2$ was activated very easily on catalytic surfaces. The mass transfer limitation of Pt–$MoO_x$/$Mo_2N$ was excluded to ensure the accuracy of kinetic data (Supplementary Fig. 14).

## Surface structure of catalyst under the RWGS reaction conditions

The deposition of Pt greatly improved the activity of catalyst, so it was important to reveal the local structure and electronic structure of Pt species in the actual reaction process. Because the very low loading of Pt (0.5 wt.%) in 0.5Pt–$MoO_x$/$Mo_2N$ could cause the poor ratio of signal to noise, we conducted the XAFS measurements over 1Pt–$MoO_x$/$Mo_2N$ (1.0 wt.%). During the $H_2$ pretreatment, the XANES profile at the Pt L₃ edge revealed a gradual decrease of the white line (feature around 11570 eV) in the first 1 h (Fig. 3a), indicating that highly oxidized Pt species were gradually reduced. However, after $H_2$ pretreatment for 2 h, the Pt could still maintain partially oxidized state. Besides, in the subsequent RWGS reaction, the partially oxidized state of Pt remained stable (Fig. 3b), suggesting the strong electron interaction between Pt and the support. The average coordination number of Pt–Pt bonds in the reaction process was ~2.5, indicating that the Pt was highly dispersed on the support in the form of clusters. Meanwhile, there was a Pt–O contribution with an average coordination number of 1.8, which originated from the bonding interaction between Pt and $MoO_x$ (Fig. 3c). Because the catalytic properties of 1Pt–$MoO_x$/$Mo_2N$ were

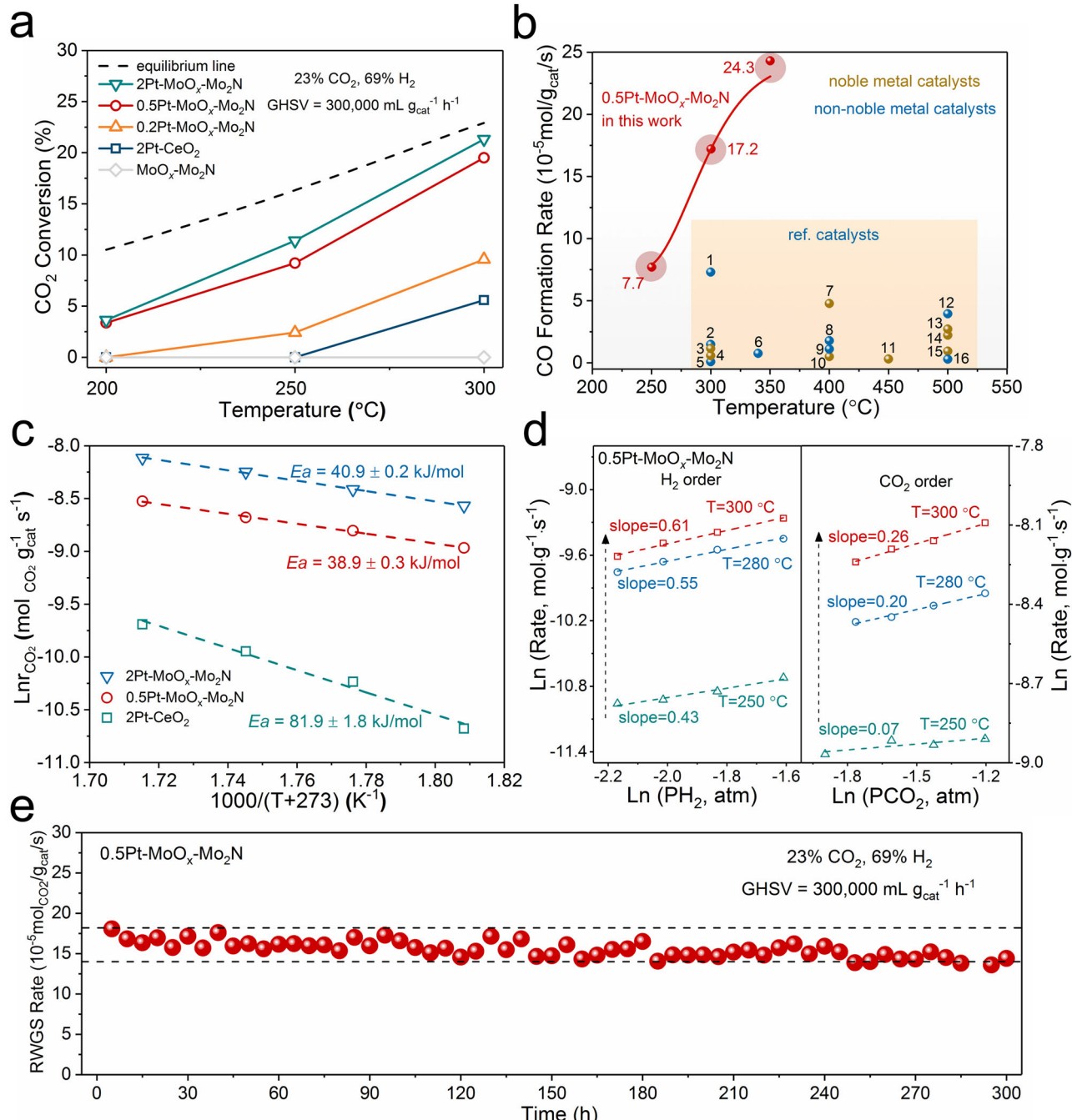

**Fig. 2 | Catalytic performance of 0.5Pt−MoO$_x$/Mo$_2$N. a** CO$_2$ conversion over different catalysts at various temperatures. **b** The mass specific activities of 0.5Pt−MoO$_x$/Mo$_2$N and other typical catalysts that were used in RWGS reaction. (1) Cu/β-Mo$_2$C (ref. 36.); (2) Cu−Zn−Al (ref. 36.); (3) Pt/CeO$_2$ (ref. 37.); (4) AuMo/SiO$_2$ (ref. 38.); (5) In$_2$O$_3$−CeO$_2$ (ref. 39.); (6) NiAu/SiO$_2$ (ref. 40.); (7) Pt-CeO$_2$ (ref. 41.); (8) TiO$_2$/Cu (ref. 42.); (9) SiO$_2$/Cu (ref. 42.); (10) Pt-TiO$_2$ (ref. 43.); (11) Rh@S-1 (ref. 44.);

(12) Ni-in-Cu (ref. 45.); (13) NiAu/SiO$_2$ (ref. 40.); (14) K$_{80}$−Pt−L (ref. 46.); (15) K$_{200}$−Pt−L (ref. 46.); (16) In$_2$O$_3$−CeO$_2$ (ref. 39.). **c** Apparent activation energy ($E_a$) of different catalysts. **d** Kinetic orders of reactants (CO$_2$ and H$_2$) for 0.5Pt−MoO$_x$−Mo$_2$N at various temperatures. **e** Long-term stability test of 0.5Pt−MoO$_x$−Mo$_2$N.

similar to those of the 0.5Pt−MoO$_x$/Mo$_2$N sample, it could be inferred that Pt in 0.5Pt−MoO$_x$/Mo$_2$N existed as clusters with smaller size, which was in good agreement with the HAADF images of the used catalyst (Fig. 1c and Supplementary Fig. 3).

Raman spectra clearly exhibited the surface structure of support in the RWGS reaction. From the ex situ Raman results of the fresh and used catalysts (Supplementary Fig. 7), there was a thin layer of MoO$_3$ on the catalyst surface. In order to explore the surface structure of support under the actual RWGS reaction conditions, systematic in situ Raman spectra were studied. As shown in Supplementary Fig. 15a,

when 0.5Pt−MoO$_x$/Mo$_2$N was treated by 5% H$_2$/Ar at room temperature (RT), MoO$_3$ was transformed into MoO$_x$ in the reduction process. The loss of oxygen implied the generation of abundant surface oxygen vacancies. With the rising of test temperature under 5% H$_2$/Ar, the characteristic peaks of MoO$_x$ remained stable. Furthermore, the in situ Raman spectra of 0.5Pt−MoO$_x$/Mo$_2$N under the RWGS reaction were measured. The surface structure of catalyst was MoO$_x$ under the actual reaction process (Supplementary Fig. 15b). Further, during the 10 h RWGS reaction, the signal of MoO$_x$ had no obvious change, suggesting that the MoO$_x$ was very stable (Supplementary Fig. 16). For the γ-Mo$_2$N

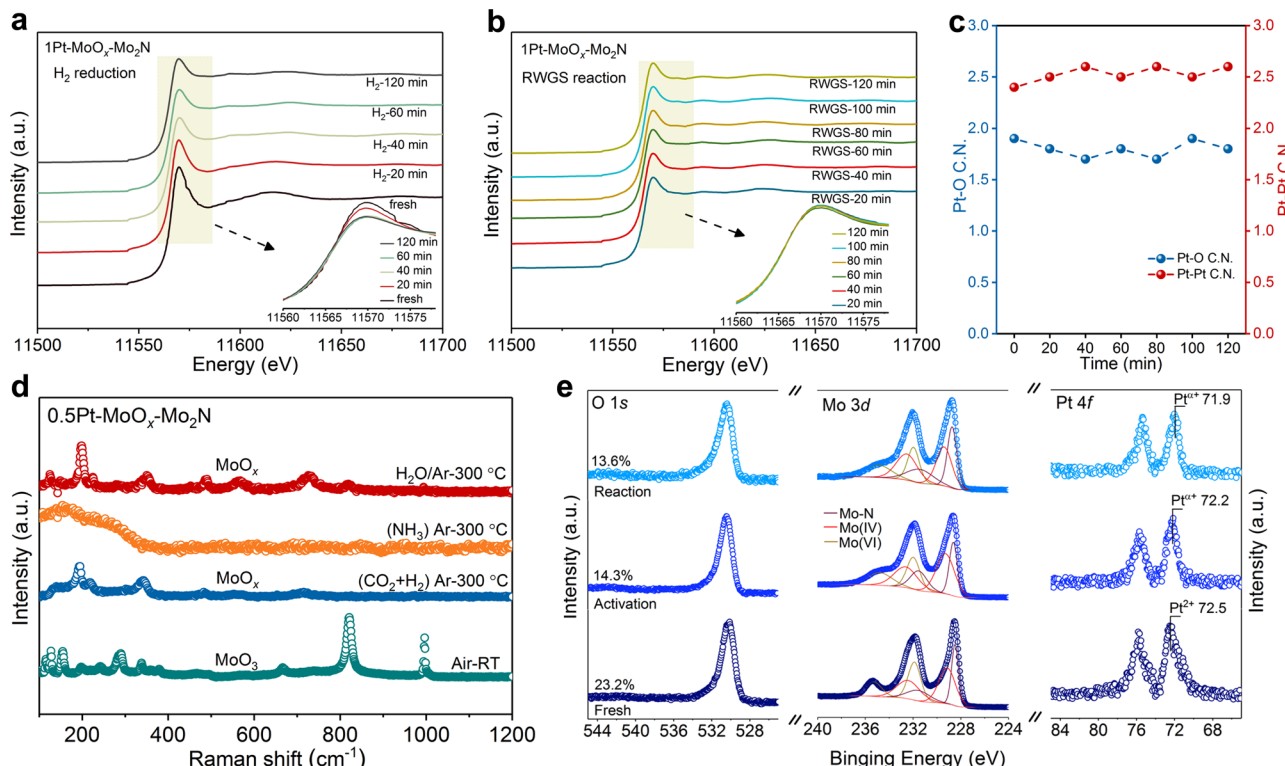

**Fig. 3 | Local coordination structure and surface structure of Pt–MoO$_x$/Mo$_2$N under H$_2$ pretreatment and RWGS reaction.** In situ XANES profile at Pt L$_3$ edge under **a** H$_2$ pretreatment and **b** RWGS reaction, respectively. **c** The change of Pt coordination number during the RWGS reaction. **d** In situ Raman spectra of 0.5Pt–MoO$_x$/Mo$_2$N under various conditions. **e** XPS spectra of fresh 0.5Pt–MoO$_3$/Mo$_2$N and quasi in situ XPS spectra of 0.5Pt–MoO$_x$/Mo$_2$N after the treatment of 5% H$_2$/Ar and 23% CO$_2$/69% H$_2$/Ar at 300 °C for 30 min.

support, the in situ Raman result (Supplementary Fig. 17) indicated that without the assistance of Pt species, the MoO$_3$ surface on the γ-Mo$_2$N was easily reduced to form MoO$_x$. However, for pure MoO$_3$, there was no signal of MoO$_x$ in the RWGS reaction process (Supplementary Fig. 18a), suggesting that the reduction of MoO$_3$ occurred only at the surface of the heterostructured MoO$_3$/γ-Mo$_2$N. The stress of MoO$_3$/γ-Mo$_2$N heterostructure was the key to reduce MoO$_3$. The MoO$_x$ with plenty of oxygen vacancies on the Mo$_2$N surface undoubtedly provided a guarantee for the formation of sufficient Pt$_n$-O$_v$ sites.

We further used quasi in situ XPS spectra to get the information on the surface structure transformation of Pt–MoO$_x$/Mo$_2$N under various atmospheres (Fig. 3e). In all XPS spectra under different atmospheres, the Mo 3d spectra could be adapted to three doublets. The predominant doublet at 228.8 and 231.9 eV was attributed to the Mo–N bond[29]. The peaks at 229.4 and 232.6 eV were ascribed to Mo$^{4+}$. The other doublet at 232.1 and 234.9 eV was attributed to Mo$^{6+}$. Compared with the fresh catalyst, after the pretreatment and the RWGS reaction, the content of oxygen atoms on catalyst surfaces decreased from 23.2 to 14.3% and 13.6%, respectively, indicating that ~40 % of oxygen atoms were reduced to form vacancies. As shown in Fig. 3d, after the treatment by NH$_3$, the Raman peaks of MoO$_x$ existed in the RWGS reaction process disappeared, which suggested the nitridation of MoO$_x$ structure. After that, under the effect of H$_2$O, the MoO$_x$ structure was regenerated. However, as shown in Supplementary Fig. 19, the broken MoO$_x$ structure could not be regenerated in the RWGS reaction, indicating that the H$_2$O generated in the RWGS reaction could leave the catalyst surface quickly without oxidizing the catalyst surface. Besides, the color change of the MoO$_3$ sample treated with NH$_3$ flow and the quasi in situ XPS spectra of 0.5Pt–MoO$_x$/Mo$_2$N (Supplementary Figs. 20 and 21) further confirmed that the NH$_3$ flow converted a part of the oxide into Mo$_2$N. Furthermore, when the catalyst was treated by NH$_3$ at 650 °C for 30 min (Supplementary Fig. 21b),

the catalyst surface was not fully nitrided into Mo$_2$N. Thus, it was difficult to completely remove the Mo-O structure on the catalyst surface. The above series of in situ characterizations showed that the catalyst surface was Pt–MoO$_x$ under the RWGS reaction.

## Pt$_n$–O$_v$ synergistic effect

We have found that the nitrogenation of the MoO$_x$ by NH$_3$ undoubtedly caused the catalyst deactivation (Fig. 4a). Meanwhile, the MoO$_x$/γ-Mo$_2$N sample without Pt deposition exhibited no activity even at 300 °C (Fig. 2a). So the excellent activity for the RWGS reaction was the result of the synergistic effect of oxygen vacancies and Pt clusters. CO$_2$ activation was a prerequisite for CO$_2$ hydrogenation. Hence, it was of great significance to explore how CO$_2$ was activated. The CO$_2$-TPD result demonstrated that the addition of Pt effectively enhanced the adsorption of CO$_2$ (Fig. 4b). Theoretical simulations showed that the CO$_2$ molecule exhibited obvious activation upon the adsorption on the Pt$_4$-MoO$_x$ surface model, which was reflected in the bending of molecular configuration and the acquisition of electric charge[30,31]. In Fig. 4c and Supplementary Fig. 22a, b, we presented three adsorption structures of CO$_2$ (denoted as CO$_2$/Pt$_4$–MoO$_x$-I, CO$_2$/Pt$_4$–MoO$_x$-II, and CO$_2$/Pt$_4$–MoO$_x$-III, respectively), whose energy were nearly degenerated with an energy difference of less than 0.1 eV. Among the three of them, the CO$_2$ molecule in CO$_2$/Pt$_4$–MoO$_x$-I had the largest angular bending (from 180.0° to 118.8° vs to 133.3° and 148.5°) and the maximum charge transfer (−0.63 e vs −0.53 e and −0.22 e), indicating that it was mostly activated. Thus, we used CO$_2$/Pt$_4$–MoO$_x$-I as a representative configuration for subsequent analysis. Projected electronic density of states (Fig. 4d) revealed that CO$_2$ had effective interactions with surrounding atoms, especially Pt-1 and Pt-4 (Fig. 4c), and the corresponding differential charge density was shown in Fig. 4e. For comparison, the adsorption of CO$_2$ was weak on both MoO$_x$ (without Pt) and Pt$_4$–MoO$_3$ (without oxygen vacancies) surface models, and the adsorbed CO$_2$

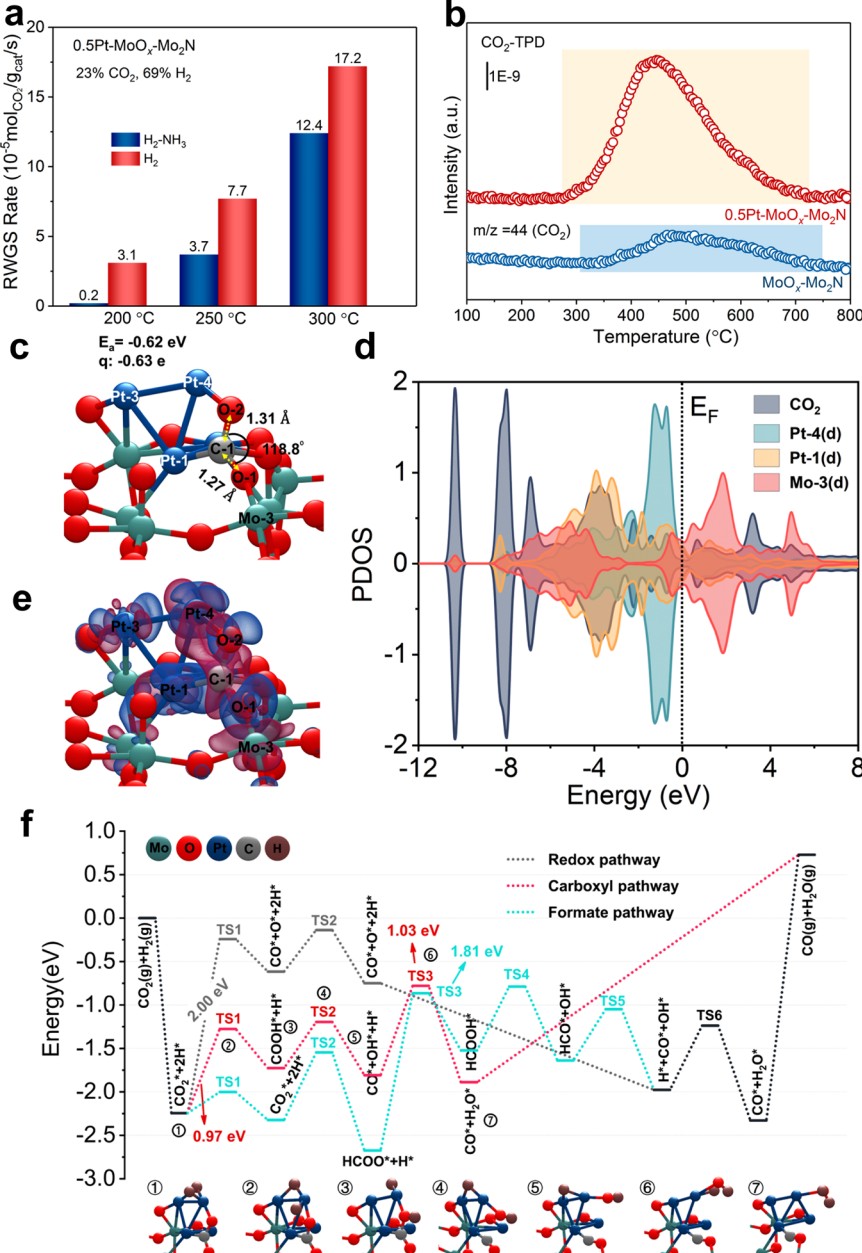

**Fig. 4 | Pt$_n$-O$_v$ synergistic effect and the proposed reaction pathways for the RWGS reaction. a** The RWGS reaction rate of 0.5Pt–MoO$_x$/Mo$_2$N after different pretreatments. **b** CO$_2$-TPD of MoO$_x$/Mo$_2$N and 0.5Pt–MoO$_x$/Mo$_2$N. **c** Adsorption structure of CO$_2$ on the Pt$_4$–MoO$_x$ surface model (denoted as CO$_2$/Pt$_4$–MoO$_x$-I). The other two adsorption configurations (denoted as CO$_2$/Pt$_4$–MoO$_x$-II and CO$_2$/Pt$_4$–MoO$_x$-III) are shown in Supplementary Fig. 22a, b. **d** Projected electronic density of states (PDOS) of the CO$_2$ adsorbate, $d$-orbitals of the two Pt atoms (Pt-1 and Pt-4), and $d$-orbitals of the Mo-3 atom in CO$_2$/Pt$_4$–MoO$_x$-I. **e** Electron transfer between adsorbed CO$_2$ and Pt$_4$–MoO$_x$ surface according to the differential charge density analysis. The blue and red colors represent the electron accumulation and depletion, respectively. **f** Energy profiles of the three reaction routes (redox, carboxyl, and formate) on Pt$_4$–MoO$_x$, depicted in gray, red, and cyan, respectively. The black line represents the common parts of the three pathways. The configurations of intermediates and transition states (TS) in the carboxyl route (energetically the most favorable according to the simulations) are displayed at the bottom, while those in the other two pathways (redox and formate) are shown in Supplementary Figs. 25 and 26, respectively.

molecule exhibited a linear configuration (Supplementary Fig. 22c, d). Hence, the coexistence of Pt and oxygen vacancies was very important for the activation of the CO$_2$ reactant, consistent with the observed poor activity of the MoO$_x$–Mo$_2$N catalyst and also that of the 0.5Pt–MoO$_x$/Mo$_2$N catalyst pretreated by oxygen (Supplementary Fig. 23). Both experimental and simulation results indicated that the Pt$_n$-O$_v$ synergistic effect promoted the catalytic performance in the RWGS reaction.

We further explored the reaction mechanism and the influence of the synergistic effect between oxygen vacancies and Pt clusters on

reaction pathways. For the RWGS reaction, the reaction mechanisms have been divided into two types, redox mechanism, and associative mechanism[32]. The difference between these two mechanisms lies in whether or not the dissociated H was involved in the formation of active intermediates[33]. The dissociation experiment of CO$_2$ indicated that CO$_2$ itself could not dissociate to CO without the assistance of H$_2$ (Supplementary Fig. 24a). And as shown in Supplementary Fig. 24b, under the effect of H$_2$, the signal intensity of CO$_2$ decreased from ~150 °C, accompanied with the formation of CO, suggesting that H$_2$ was necessary for the reaction process[34]. A similar result was obtained

by the in situ diffused reflectance infrared Fourier transform spectroscopy (DRIFTS) spectra. As illustrated in Supplementary Fig. 24c, when the catalyst was treated with $CO_2$ at 300 °C, no CO gaseous signal appeared. However, in the presence of $CO_2$ and $H_2$, the distinct gaseous signal of CO was observed (Supplementary Fig. 24d), indicating that $H_2$ was involved in the formation of active intermediates. Thus, the reaction catalyzed by 0.5Pt–$MoO_x$/$Mo_2N$ followed the associative mechanism. Computational simulations were performed to investigate reaction pathways, as shown in Fig. 4f. Here, both the redox and the associative (including the carboxyl and the formate routes) mechanisms were considered (Fig. 4f). The redox mechanism would not appear in the reaction process (Supplementary Fig. 25), since the corresponding highest energy barrier (2.00 eV in TS1) was much higher than that of the associative mechanism (1.03 eV in TS3 from the carboxyl route). This was consistent with the experiments that $CO_2$ could not dissociate by itself. Regarding the associative mechanism, it was found that Pt atoms and oxygen vacancies were both involved in the reactions. The adsorption and activation of $CO_2$ was the prerequisite for the processing of RWGS reaction. Computational simulations indicated that Pt cluster or oxygen vacancy alone could not effectively adsorb and activate $CO_2$ (Supplementary Fig. 22c, d). However, $CO_2$ could be easily activated at the interface between Pt cluster and oxygen vacancy (Fig. 4c and Supplementary Fig. 22a, b). The activation of $CO_2$ at the interface between Pt cluster and oxygen vacancy guaranteed the further formation of reactive intermediate. Besides, the intermediates in the reaction pathways, like COOH* and HCOO*, were connected with Pt and an adjacent unsaturated Mo atom that appeared upon the removal of oxygen atoms. These results further supported the synergistic effect of Pt clusters and oxygen vacancies in the catalytic process. Our calculations showed that the energy barrier in the carboxyl route (1.03 eV in TS3) was lower than the value (1.81 eV in TS3) in the formate one, suggesting that the carboxyl route was energetically more favorable (Fig. 4f and Supplementary Fig. 26). We noted in passing that from our experiments, we could not determine whether the actual reaction process was via the carboxyl route or the formate pathway (But no matter which of the two was adopted, the cooperation of Pt clusters and oxygen vacancy would always play a crucial role in promoting the RWGS reaction).

## Discussion

Synergizing multiple catalytic sites is the key to enhance the efficiency of catalysts in heterogeneous catalysis. In this work, using the stress of $MoO_3$/γ-$Mo_2N$ structure and the interaction between Pt and support, efficient catalysts with high density of $Pt_n$–$O_v$ synergistic sites were successfully constructed. The Pt–$MoO_x$/$Mo_2N$ catalyst showed excellent activity and solid stability in low-temperature RWGS reactions, which was significantly better than other reported catalysts. During the pretreatment and reaction process, surface $MoO_3$ was transformed into $MoO_x$ with the creation of a large number of oxygen vacancies. Through in situ reduction, isolated Pt atoms were converted to Pt clusters. Highly dispersed Pt clusters were stably bonded with the O atoms of $MoO_x$, in which the interaction between Pt and $MoO_x$ prevented the Pt clusters from sintering. By the combination of experimental investigations and theoretical calculations, the synergistic catalytic effect between Pt clusters and oxygen vacancies was proved to be the key to promote the reaction. This work provides an example to explore the interfacial structure of catalysts in the actual reaction process, and paves a new way to amplify the synergistic effect of active metal and support by taking the advantages of heterostructure.

## Methods
### Preparation of $MoO_3$
The $MoO_3$ Precursor was prepared by hydrothermal method. Firstly, 1 g of P123 was dissolved by 40 mL deionized water in a Teflon bottle. Then, 0.9 g of $Na_2MoO_4 \cdot H_2O$ and 5 mL deionized water were added

onto the above suspension to form a stock solution. Next, 3 mL of concentrated hydrochloric acid (37 wt.%) was added in drops to the stock solution. Finally, the Teflon bottle was sealed into a stainless autoclave tightly and heated at 100 °C for 12 h in the oven. The resulting precipitates are collected by centrifugation and washed with deionized water and absolute ethanol, followed by drying in the oven at 70 °C for 10 h. The resulting solid is heated in tube furnace at 400 °C for 4 h to obtain the $MoO_3$ powder. For the preparation of γ-$Mo_2N$ support, the above $MoO_3$ powder was ground and transferred into a quartz tube and ammonized by pure $NH_3$ (40 mL/min) at 650 °C for 4 h. After cooling, the prepared γ-$Mo_2N$ was passivated with 1% $O_2$/Ar mixed gas for 2 h. The 1% $O_2$/Ar mixed gas could oxide the surface of the fresh γ-$Mo_2N$ to prevent the full oxidation of the bulk-phase γ-$Mo_2N$.

### Preparation of Pt–$MoO_3$/$Mo_2N$ catalysts
A series of Pt–$MoO_3$–$Mo_2N$ catalysts were prepared by the ethylene glycol method. Firstly, the γ-$Mo_2N$ support (400 mg) and a designed amount of chloroplatinic acid (0.19 mol/L, 54 μL) were added into a beaker containing 100 mL ethylene glycol, and then stirred for 0.5 h and ultrasonicated for another one hour to achieve good dispersion of the slurry. Next, adjust the pH of the slurry to 1.5 with HCl aqueous solution (6.2 wt. %). After that, the slurry was refluxed at 140 °C for 2 h. The resulting precipitates are collected by centrifugation and washed with deionized water and absolute ethyl alcohol, followed by drying in the oven at 40 °C to obtain the fresh catalysts. In this work, the obtained catalysts were nominated as xPt–$MoO_3$/$Mo_2N$ (x = 0.2, 0.5, 1 and 2), where x is the platinum content in weight percent (x = [Pt/$Mo_2N$] $_{wt}$× 100%).

### Preparation of reference Catalyst (Pt–$CeO_2$)
The Pt–$CeO_2$ catalyst was synthesized by deposition–precipitation method, according to the previous reports[35]. 0.5 g $CeO_2$ nanorod support was dispersed into 30 mL ultrapure water. A certain amount of chloroplatinic acid solution (0.19 mol/L, 68 μL) was then added to the obtained suspension drop by drop and the pH of the solution was kept at 9 with the assistance of sodium carbonate solution (0.1 mol/L) during the whole process. The resulting precipitate was aged at room temperature for 1 h, and washed by ultrapure water. The solid was then dried at 70 °C for 10 h in the oven and calcined at 400 °C for 4 h in muffle furnace.

### X–ray diffraction (XRD)
For the ex situ XRD data, all experiments were operated on PANalytical X'pert3 powder diffractometer with CuK radiation (λ = 0.15406 nm).

### Transmission electron microscope (TEM)
TEM was conducted by JEM-2100F (JEOL) instrument operating at 200 kV. The samples were sonicated in ethanol and being dropped on the carbon-coated Cu grid before test. The High-angle annular dark-field scanning transmission electron microscopy (HAADF-STEM) images were obtained on a Thermo Scientific Themis Z microscope equipped with a probe-forming spherical-aberration corrector.

### Inductively coupled plasma-atomic emission spectroscopy (ICP-AES)
For 0.5Pt–$MoO_3$/$Mo_2N$ catalyst, the ICP-AES measurement was carried out on an IRIS Intrepid II XSP instrument (Thermo Electron Corporation).

### X-ray photoelectron spectroscopy (XPS)
Quasi in situ XPS experiments were carried out on a Thermo Scientific ESCALAB Xi⁺ XPS instrument. The spectrums of Mo 3d, C 1s, O 1s, and Pt 4f were obtained after the catalysts were pretreated by different atmospheres and temperatures for 1 h.

## Raman spectroscopy

Ex situ and in situ Raman spectra were acquired on a Raman microscope system (HORIBA JY) with laser excitation at 473 nm. The integration times of Ex situ and in situ Raman spectra were 60 s and 5 min, respectively. For the in situ Raman, the micro-Raman reaction cell (Xiamen TOPS) equipped with a quartz window has a heating module that controls the test temperature. Five kinds of programs were conducted as follows.

(i) The $0.5Pt–MoO_x/Mo_2N$ catalyst was pretreated by 5% $H_2$/Ar at 300 °C for 30 min. After cooling to room temperature, the gas flow was switched to RWGS reaction gas (33% $CO_2$, 67% $H_2$), and then heated to 300 °C for 10 h. The Raman spectra was collected at 100, 200, and 300 °C, respectively.

(ii) Firstly, the $0.5Pt–MoO_x–Mo_2N$ catalyst was treated with 5% H2/Ar at 300 °C for 30 min. Then, using the pure NH3 to purge the sample from room temperature and heated to 300 °C for 30 min. After that, the gas flow was changed to RWGS reaction gas (25% $CO_2$, 75% H2) to treat the sample at 300 °C for 120 min. Finally, use ~3% $H_2O$/Ar to purge the catalyst for another 30 min at 300 °C. The Raman spectra were collected under RWGS reaction gas, NH3, $H_2O$/Ar at 300 °C, respectively.

(iii) The whole test process was the same as the procedure in (iii), except that the RWGS reaction gas (25% $CO_2$, 75% $H_2$) and pure $NH_3$ were switched in a different order.

## $CO_2$ temperature-programmed desorption ($CO_2$-TPD), $CO_2$ dissociation experiment, temperature-programmed surface reaction (TPSR)

All these experiments were performed on a lab-made reactor and the outlet gases were analyzed by mass spectrum (LC-D200M, TILON). For these different three types of tests, the catalysts were firstly reduced in 5% $H_2$/Ar at 300 °C for 30 min, and then flushed with Ar gas flow (30 mL min$^{-1}$) at room temperature for 30 min. For $CO_2$-TPD, the catalysts were saturated with 2% $CO_2$/Ar (30 mL min$^{-1}$) at room temperature for 30 min followed by purging with Ar gas flow (30 ml min$^{-1}$) for 30 min to purge all the physically adsorbed $CO_2$ molecules on the surface of catalysts and the residual $CO_2$ in the reaction tube. And then the $CO_2$-TPD experiment was measured from room temperature to 300 °C with a ramping rate of 10 °C min$^{-1}$ under Ar gas flow (30 mL min$^{-1}$). For $CO_2$ dissociation experiment, after $H_2$ activation and Ar gas purge, the samples were purged with 2% $CO_2$/Ar with heating from room temperature to 300 °C. For TPSR, the samples were treated with mixed gas with 15% $CO_2$ and 30% $H_2$ with heating from room temperature to 300 °C.

## In situ diffuse reflectance infrared Fourier transform spectroscopy (DRIFTS)

All of the in situ DRIFTS spectra were collected by using a Bruker Vertex 70 FTIR spectrometer with a mercury cadmium telluride (MCT) detector cooled with liquid nitrogen. The treatment process of $CO_2$ on the $0.5Pt–MoO_x–Mo_2N$ catalyst was investigated by in situ DRIFTS measurement at 300 °C. Prior to the in situ DRIFTS measurement, ~30 mg sample was pretreated at 300 °C for 30 min under 5% $H_2$/Ar mixed gas. The background spectra were collected under $N_2$ atmosphere at 4 cm$^{-1}$ resolution at 300 °C. After the collection of the background spectrum, the mixed gas consisted of 2% $CO_2$/Ar and was introduced into the chamber. Continuous recording of the IR profiles was maintained for 5 min. As for the test under RWGS reaction conditions, after background acquisition, the reaction gas with 15% $CO_2$/ 30% $H_2$/55% $N_2$ is introduced into the in situ chamber. All DRIFTS results were analyzed by using OPUS software.

## XAFS (X-ray absorption fine spectroscopy) experiments

XAFS was performed at the BL11B beamline in Shanghai Synchrotron Radiation Facility (SSRF, Energy 3.5 GeV, Current 250 mA in maximum,

Si (111) double-crystals as double crystal monochromator which could cover the photon energy range from 4.5 to 18 KeV). The samples were measured in fluorescence mode, using a Lytle detector to collect the data.

XAFS of $H_2$ reduction process: The XAFS sample was sealed in the reaction cell. The catalyst was pretreated by 5% $H_2$/He at 300 °C for 2 h. The XAFS was continuously collected during the reduction process.

XAFS of the RWGS process: After the sample was reduced, the XAFS was collected under the reaction conditions.

All XAFS spectra were analyzed using the Ifeffit package version 1.2.11.

## Catalytic measurements and kinetic tests

The activity of the catalysts for RWGS reaction was conducted by a fixed-bed reactor at atmospheric pressure. For temperature-dependant activity test, 10 mg sieved sample (20–40 mush) was mixed with 90 mg inert $SiO_2$ and was packed into a quartz tube. Before the RWGS reaction, the sample was pretreated by 5% $H_2$/Ar (30 mL/min) at 300 °C for 30 min. And then when the temperature of the catalyst dropped to room temperature, the gas flow was switched into RWGS reaction gas flow (23% $CO_2$, 69% $H_2$, 8% $N_2$) with 50 mL/min. At each test temperature, the product was analyzed after 60 min of steady-state reaction. For the test of reaction rate, in order to obtain reaction rates in the kinetics region, appropriate amount of catalysts diluted with $SiO_2$ were used and the $CO_2$ conversion rate was controlled at a relative low level (<20%) by changing gas flow rate. The outlet product was analyzed by an on-line gas chromatograph equipped with a thermal conductivity detector (TCD). The gas flow rate was determined by the inner standard method, in which the $N_2$ was used as the inner standard. The $CO_2$ conversion and CO selectivity were calculated by the following equations:

$$X_{CO_2}(\%) = \frac{n_{CO_2}^{in} - n_{CO_2}^{out}}{n_{CO_2}^{in}} \times 100\% = \left(1 - \frac{A_{CO_2}^{out}/A_{N_2}^{out}}{A_{CO_2}^{in}/A_{N_2}^{in}}\right) \times 100\% \quad (1)$$

where $n_{CO_2}^{in}$ is the concentration of $CO_2$ in the reaction stream, and $n_{CO_2}^{out}$ is the concentration of $CO_2$ in the outlet gas. $A_{CO_2}^{in}$ and $A_{N_2}^{in}$ refer to the chromatographic peak area of $CO_2$ and $N_2$ in the inlet gas, respectively, and $A_{CO_2}^{out}$ and $A_{N_2}^{out}$ refer to the chromatographic peak area of $CO_2$ and $N_2$ in the outlet gas, respectively. The chromatographic peak area of each component is proportional to the concentration of each component.

The selectivity of CO was calculated as

$$S_{CO}(\%) = \frac{n_{CO}^{out}}{n_{CO}^{out} + n_{CH_4}^{out}} \times 100\% = \frac{A_{CO}^{out} \times f_{CO/N_2}}{A_{CO}^{out} \times f_{CO/N_2} + A_{CH_4}^{out} \times f_{CH_4/N_2}} \times 100\%$$

$$(2)$$

where $n_{CO}^{in}$ and $n_{CH_4}^{in}$ refer to the concentration of CO and $CH_4$ in the outlet gas, respectively. $f_{CO/N_2}$ and $f_{CH_4/N_2}$ are relative correction factors of CO to $N_2$ and $CH_4$ to $N_2$, respectively, which are determined by the calibrating gas. $A_{CO}^{out}$ and $A_{CH_4}^{out}$ are the chromatographic peak area of CO and $CH_4$ detected by the TCD in the outlet gas.

The carbon balance was calculated as

$$C_{balance}(\%) = \frac{(A_{CO_2}^{out} \times f_{CO_2/N_2} + A_{CO}^{out} \times f_{CO/N_2} + A_{CH_4}^{out} \times f_{CH_4/N_2}) \times (1 - X_{CO_2})}{A_{CO_2}^{out} \times f_{CO_2/N_2}} \times 100\%$$

$$(3)$$

where $f_{CO_2/N_2}$ is relative correction factor of $CO_2$ to $N_2$, which is determined by the calibrating gas. $A_{CO_2}^{out}$ is the chromatographic peak area of $CO_2$ detected by the TCD in the outlet gas.

## Apparent activation energy ($E_a$) and apparent kinetic orders

The reactor used for the $E_a$ test is the same as that used for the activity evaluation. The appropriate amount of catalysts diluted with $SiO_2$

powder were used in the kinetics experiments. During kinetics tests, the $CO_2$ conversion is controlled below 15% by adjusting the temperature and gas flow rate. The reaction orders of $CO_2$ and $H_2$ for the catalysts were collected under 250, 280, and 300 °C. The RWGS activity was recorded while the concentration of $CO_2$ or $H_2$ in the reaction gas was varied on purpose.

## Theoretical calculations
Details of the computational methods and the simulation model are put in the Supplementary Information.

## Data availability
The main data supporting the findings of this study are available within the article and its Supplementary Information. Extra data are available from the corresponding author upon request. Source data are provided with this paper.

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

## Acknowledgements
This work was financially supported from the National Key Basic Research Program of China (2021YFA1501102, 2021YFA1501103), the National Science Foundation of China (no. 21803036, 22075166, 21725301, 21932002), the Taishan Scholar Project of Shandong Province of China, the CAS Project for Young Scientists in Basic Research (YSBR-054), and the Young Scholars Program of Shandong University (2018WLJH49). We thank the Center of Structural Characterizations and Property Measurements at Shandong University for the help on sample characterizations. D.M. acknowledges support from the Tencent Foundation through the XPLORER PRIZE.

## Author contributions
C.J.J. and D.M. supervised the work; H.X.L., C.J.J., and D.M. designed the experiments, analyzed the results, and wrote the manuscript; J.Y.L. and Q.F. carried out the first-principles simulations; X.T.Q performed the XAFS measurements. H.X.L., K.X., and W.W.W. performed the in situ DRIFTS, in situ Raman, and Quasi in situ XPS; H.X.L., K.X., and H.Y. performed the catalysts preparation, catalytic tests, and the TPR tests; C.M. performed the aberration-corrected HAADF-STEM measurements and analyzed the results. D.Q.X. gave a lot of suggestions and helped to modify the manuscript.

## Competing interests
The authors declare no competing interests.
