## [Peer Review File · Nature Communications]

Title: Ptn-Ov synergistic sites on MoO_x/γ-Mo₂N heterostructure for low-temperature reverse water–gas shift reactionREVIEWER COMMENTS

Reviewer #1 (Remarks to the Author):

Recommendation: Publish after major revisions noted.

Comments:

This manuscript by Liu et al., reports a study of the evolution and dynamics of the chemical, structural, and electronic properties at the Pt/ MoO_x/Mo₂N heterostructure interface and its consequences in low-temperature RWGS reaction. The authors have characterized the structural dynamics by spectroscopic and microscopic analysis using multiple operando XPS, XAS, Raman, DRIFTS and conventional HRTEM-STEM techniques. They found an interesting experimental result that demonstrate the importance of oxygen vacancies and the stabilization of Pt clusters related to the vacancies. This manuscript contains relevant information and might be published in Nature Communications after addressing the following issues:

1. The authors suggested that reaction takes place following an associative mechanism against redox mechanism. However, the role of vacancies on the associative mechanism is poorly discussed. The authors might like to have more discussion on it.
2. It is well-known that RWGS pathway for CO formation is often thermodynamic favored at high temperatures while methanation is favored at low temperature. On the other side, although the reaction was performed at ambient pressure, it cannot be ruled out the formation of other oxygenated species such as formic acid or methanol. The presence of other by-products must be discussed.
3. By XAS measurements the authors observed that Pt species was not totally reduced to metallic Pt after reduction process. Apparently, the authors suggested that Pt species remains stable during RWGS reaction. What is the dynamic of the chemical state of Pt species and the surface restructuration during RWGS reaction?
4. The conversion and selectivity calculations use the flow rates of chemical species, which is not directly measured but must be calculated from concentration (I assume, since the method of gas composition detection has not been described). What are the equations used to convert concentration of species to their flow rates? These must be presented in the methods section. Is there an internal standard? Have carbon balances been calculated?

Minor comments:

The units of apparent activation energy must be revised: KJ mol⁻¹ by kJ mol⁻¹

Reviewer #2 (Remarks to the Author):

The authors have submitted an interesting manuscript on a new catalyst for the RWGS reaction, based on platinum on Mo₂N that forms an MoO_x surface layer. This way an unusually high concentration of surface oxygen vacancies are formed that cooperate with the redox-active platinum sites.

The work is of high significance for the field because the presented catalyst excels the catalysts presented in literature so far by a much higher activity. Thermodynamic equilibrium conversion can be achieved at 300°C already. The conclusion are sufficiently supported by the provided data to a large extent.

It can be expected that the insights provided by the authors could be helpful for the scientific community to better understand the principles of active and selective RWGS catalysts. The practical impact of the work cannot be assessed, since the long-term stability of the catalyst is unclear, the possibility to prepare a practically applicable catalyst remains open and the process economics for a catalyst working at low reaction temperatures with very limited equilibrium conversion to the desired products might be unfavorable. However, the provided scientific insights could justify publication in Nature Communications.

There are a number of questions and comments that should be addressed before publication can be recommended:

page 3, top: "...efficient supported catalysts always require synergy between multiple components^{1–4}." This is daring statement and should be better justified. "Synergy" means that all components profit from each other. However, in many cases of supported catalysts only the catalytically active component profits from the support, which cannot be called "synergy".

page 3, middle: "the flexibility of surface oxygen atoms" What do the authors mean by "flexibility"? This term could mean almost everything and needs specification.

page 3, bottom: "due to their unique electronic structure⁹." Please specify what is "unique".

page 3, bottom: The term "deoxidize" is used several times in the manuscript. This term is unknown to me. I would call this "reduce".

page 4, top: "However, CO₂ molecule is very stable, and the RWGS is endothermic, making the conversion of CO₂ at low temperature very challenging." I wonder how attractive low-temperature RWGS is, since the equilibrium is shifted to the reactants at low temperatures, resulting in low maximum conversions. Maybe more stable RWGS catalysts at higher temperatures are more attractive and desirable from an application point of view.

page 4, top: "...for the RWGS catalysts, active metal clusters can dissociate H₂, while the vacancies can activate CO₂. Therefore, the creation of catalysts with high-density synergistic sites between active metal cluster and oxygen vacancy may greatly improve the low-temperature activity of catalysts."

Usually, only one reaction step is the rate-determining step and the authors should comment which one determines the overall reaction rate. In this regard, the authors should explain what they mean with the vague term "synergistic sites".

Page 5, middle: "isolated Pt atoms were anchored on the γ -Mo₂N support (Figure 1b and

Supplementary Figure 2)." The isolated Pt atoms cannot be seen in the presented figures, although they are marked. Based on the shown figures, the existence of the isolated platinum atoms depend on belief. Scientific evidence has to be provided.

page 6, middle: "the single Pt atoms in 0.5Pt-MoO₃/γ-Mo₂N agglomerated into nanoclusters during the RWGS reaction" See my comment above. The agglomeration is also invisible from the presented figures. Scientific evidence has to be provided.

page 8, middle: "0.5Pt-MoO_x/γ-Mo₂N showed excellent stability after 300 h reaction". The authors should be more honest in the description of the measured data. I agree that the catalyst is relatively stable, but I still see a slight deactivation trend over 300 h.

page 11, middle: "After the sample was treated by NH₃ flow at 300 °C, the proportion of oxygen content decreased, suggesting that the NH₃ flow converted part of the oxide into Mo₂N (Figure 3d and Supplementary Figure 19–21)" It is unclear for the reader how to see this claim from Figure 3d. A more detailed argumentation is required.

page 13, bottom: "CO gaseous signal" DRIFTS provide signals of adsorbed CO. If gaseous CO was indeed measured more information should be provided. How was the DRIFTS experiments designed? Was the gas phase over the catalyst measured as well?

page 14, bottom: "We noted in passing that in experiments, which of the two routes would be taken in the reaction process was not identified." This sentence is hardly understandable. Please rephrase.

page 16, middle: "concentrated hydrochloric acid". Please specify the concentration.

page 16, bottom: "ethyl alcohol". Please write ethanol instead.

page 17, top: "designed amount of chloroplatinic acid". This is unclear and should be rephrased. Please be specific with respect to the amounts.

page 17, middle: "half an hour" = 0.5 h

page 17, middle: "ultrasonic". Should read "ultrasonicated".

page 17, middle: "HCl aqueous solution". Please specify concentration.

page 17, middle: "according to previous reports". Please add reference.

page 17, middle: "certain amount of chloroplatinic acid solution". Please specify amount and concentration.

page 17, middle: "sodium carbonate solution". Please specify concentration.

page 19, top: "clear away". Please rephrase.

page 19, top: "physical adsorbed CO₂ molecules" should read "physically adsorbed CO₂ molecules".

Supplementary Figure 9: Please clearly label the figures, e.g. what is fresh and used? What does e and f show? Moreover, the spectra are only superficially explained in the main manuscript.

page S14: "For all Pt-MoO₃/Mo₂N catalysts with different Pt loading, the stronger intensity of the Pt-Pt coordination peak of the fresh catalysts than that of the used catalysts was confirmed, suggesting the reduction and aggregation of the Pt species in the RWGS reaction." From the figure, I see that the Pt-Pt coordination peak of the used catalyst is stronger, which is in conflict with the authors statement.

Supplementary Figure 14. Labels a) and b) are mixed.

S22, bottom: "... that the surface of MoO₃ is difficult to deoxygenate during the RWGS reaction and generate a lot of oxygen vacancies." The listed consequences are in obvious conflict with each other. Please rephrase.

Supplementary Figure 22. What does "Bader charge" mean?

Supplementary Figure 23. A catalyst can be pretreated but not an activity test.

Supplementary Figure 24: This figure is unclear. What was measured under what conditions? Which intensities are given at the y-axis.

Supplementary Figures 25 and 26. There is not reference to these figures in the manuscript.

Responses to the Reviewers' Comments and the Corresponding Revisions

To Reviewer 1:

Reviewer #1: This manuscript by Liu et al., reports a study of the evolution and dynamics of the chemical, structural, and electronic properties at the Pt/MoO_x/Mo₂N heterostructure interface and its consequences in low-temperature RWGS reaction. The authors have characterized the structural dynamics by spectroscopic and microscopic analysis using multiple operando XPS, XAS, Raman, DRIFTS and conventional HRTEM-STEM techniques. They found an interesting experimental result that demonstrates the importance of oxygen vacancies and the stabilization of Pt clusters related to the vacancies. This manuscript contains relevant information and might be published in Nature Communications after addressing the following issues.

Comment 1: *The authors suggested that reaction takes place following an associative mechanism against redox mechanism. However, the role of vacancies on the associative mechanism is poorly discussed. The authors might like to have more discussion on it.*

Response: Thanks for the reviewer's valuable comments. In our work, the role of the oxygen vacancy can be reflected in the following two aspects.

(1) The oxygen vacancy directly promoted the activation of CO₂.

The adsorption and activation of CO₂ was the prerequisite for the processing of the RWGS reaction. Computational simulations indicated that Pt cluster or oxygen vacancy alone could not effectively adsorb and activate CO₂ (Figure R1c and d). However, CO₂ could be easily activated at the interface between Pt cluster and oxygen vacancy (Figure R1a and b). Therefore, oxygen vacancies were directly involved in the activation of CO₂, promoting the catalytic performance. Besides, when the surface MoO_x structure was partially nitrated by NH₃, the catalyst exhibited much lower activity (Figure R2), which again confirmed that the abundant surface oxygen vacancies was the key to the high catalytic efficiency of Pt-MoO_x/Mo₂N for the low-temperature RWGS reaction.

Figure R1. Adsorption structures of a CO₂ molecule on different surfaces (a, b, Pt₄-MoO_x surface; c, Pt₄-MoO₃ surface; d, MoO_x surface). E_a: adsorption energy of CO₂; q: calculated number of charges carried by CO₂ via the Bader charge analysis.

Figure R2. The RWGS reaction rate of 0.5Pt-MoO_x/Mo₂N after different pretreatments.

(2) The oxygen vacancy was the site for the formation of reactive intermediate.

In catalytic reaction pathway, Pt cluster worked together with the adjacent oxygen vacancy at the Pt-MoO_x interface, making the dissociated H atom to react with the activated CO₂ to form reactive intermediate. The formed intermediates in the reaction pathways, like COOH* and HCOO*, were connected with Pt and an adjacent unsaturated Mo atom that appeared upon the removal of oxygen atoms. Therefore, oxygen vacancy was also the site for the formation of reactive intermediate.

The relevant supplements have been added in the revised manuscript on page 14, line 14–24 (highlighted in yellow). Thanks for the reviewer's comment again.

Comment 2: It is well-known that RWGS pathway for CO formation is often thermodynamic favored at high temperatures while methanation is favored at low temperature. On the other side, although the reaction was performed at ambient pressure, it cannot be ruled out the formation of other oxygenated species such as formic acid or methanol. The presence of other by-products must be discussed.

Response: The reviewer's comment is highly appreciated. In addition to the detection of CO and CH₄ with a thermal conductivity detector (TCD) in chromatograph, the other hydrocarbons were further detected by using a HP-PONA capillary column with a flame ionization detector (FID). **However, over the entire test temperature range (200 °C–300 °C), no hydrocarbon was detected, indicating the very good selectivity for CO (Figure R3a).** In order to further confirm the absence of other productions such as methane, formic acid and methanol, the temperature-programmed surface reaction (TPSR) with MS as analyzer was measured. As shown in Figure R3b, during the whole reaction process, **only the signal of CO increased with the increase of reaction temperature, which again demonstrated that CO was the only detected product.** Thanks for the reviewer's valuable comments again.

Figure R3. (a) The TCD (pink line) and FID (black line) online signal of the production during the reaction at 300 °C; (b) The temperature-programmed surface reaction (TPSR) profile for 0.5Pt-MoO_x/Mo₂N. The sample was pretreated with 5% H₂/Ar at 300 °C for 30 min and then purged with Ar at room temperature to remove the adsorbed H₂. Finally, the signal evolution was collected under RWGS reaction conditions at a ramping rate of 10 °C/min.

Comment 3: By XAS measurements the authors observed that Pt species was not totally reduced to metallic Pt after reduction process. Apparently, the authors suggested that Pt species remains stable during RWGS reaction. What is the dynamic of the chemical state of Pt species and the surface restructuring during RWGS reaction?

Response: Thanks for the reviewer's valuable comments. **The chemical state of Pt species depends on the interaction between Pt species and the support.** During the RWGS reaction, the stable Pt-O bond indicated the stable interaction between Pt species and MoO_x structure, which prevented Pt species from being completely reduced and aggregated into large-size particles.

The surface structure of Pt-MoO_x/Mo₂N was elucidated from the following two aspects.

(1) The structure of Pt species in the RWGS reaction.

As shown in Figure R4a, the *in situ* XANES profile of 1Pt-MoO_x/Mo₂N suggested that Pt species could maintain stable partially oxidized state during the RWGS reaction. And from Figure R4b, during the RWGS reaction, the coordination number (~2.5) of Pt-Pt bonds was stable, indicating that the Pt species could maintain stable structure in the form of clusters.

Figure R4. (a) *In situ* XANES profile of 1Pt-MoO_x/Mo₂N at Pt L₃-edge under the RWGS reaction; (b) The coordination environment of Pt in 1Pt-MoO_x/Mo₂N during the RWGS reaction.

(2) The structure of MoO_x in the RWGS reaction.

The *in situ* Raman spectra (Figure R5) of 0.5Pt-MoO_x/Mo₂N suggested that during the RWGS reaction, the catalyst could maintain the stable MoO_x surface structure.

Based on the above *in situ* characterization results, **Pt species and surface MoO_x structure were stable in the RWGS reaction and no surface restructuring was detected. Thanks for reviewer's valuable comments again.**

Figure R5. *In situ* Raman spectra of 0.5Pt-MoO_x/Mo₂N under the RWGS reaction.

Comment 4: *The conversion and selectivity calculations use the flow rates of chemical species, which is not directly measured but must be calculated from concentration (I assume, since the method of gas composition detection has not been described). What are the equations used to convert concentration of species to their flow rates? These must be presented in the methods section. Is there an internal standard? Have carbon balances been calculated?*

Response: Thanks for the reviewer's valuable comments. In this work, a gas chromatograph (GC-9160, Ouhua, China) with a thermal conductivity detector (TCD) was used to detect gas composition, including CO₂, N₂, CH₄ and CO. **The N₂ in the flow was used as the inner standard.** The CO₂ conversion rate (X_{CO_2}) was calculated by using the following equation.

$$X_{\text{CO}_2}(\%) = \frac{n_{\text{CO}_2}^{\text{in}} - n_{\text{CO}_2}^{\text{out}}}{n_{\text{CO}_2}^{\text{in}}} \times 100\% = \left(1 - \frac{A_{\text{CO}_2}^{\text{out}}/A_{\text{N}_2}^{\text{out}}}{A_{\text{CO}_2}^{\text{in}}/A_{\text{N}_2}^{\text{in}}}\right) \times 100\% \quad (1)$$

where $n_{\text{CO}_2}^{\text{in}}$ is the concentration of CO₂ in the reaction stream, and $n_{\text{CO}_2}^{\text{out}}$ is the concentration of CO₂ in the outlet gas. $A_{\text{CO}_2}^{\text{in}}$ and $A_{\text{N}_2}^{\text{in}}$ refer to the chromatographic peak area of CO₂ and N₂ in the inlet gas, respectively, and $A_{\text{CO}_2}^{\text{out}}$ and $A_{\text{N}_2}^{\text{out}}$ refer to the chromatographic peak area of CO₂ and N₂ in the outlet gas, respectively. The

chromatographic peak area of each component is proportional to the concentration of each component.

The selectivity of CO was calculated as

$$S_{\text{CO}}(\%) = \frac{n_{\text{CO}}^{\text{out}}}{n_{\text{CO}}^{\text{out}} + n_{\text{CH}_4}^{\text{out}}} \times 100\% = \frac{A_{\text{CO}}^{\text{out}} \times f_{\text{CO}/\text{N}_2}}{A_{\text{CO}}^{\text{out}} \times f_{\text{CO}/\text{N}_2} + A_{\text{CH}_4}^{\text{out}} \times f_{\text{CH}_4/\text{N}_2}} \times 100\% \quad (2)$$

where $n_{\text{CO}}^{\text{in}}$ and $n_{\text{CH}_4}^{\text{in}}$ are the concentration of CO and CH₄ in the outlet gas, respectively. f_{CO/N_2} and $f_{\text{CH}_4/\text{N}_2}$ are relative correction factors of CO to N₂ and CH₄ to N₂, respectively, which are determined by the calibrating gas. $A_{\text{CO}}^{\text{out}}$ and $A_{\text{CH}_4}^{\text{out}}$ are the chromatographic peak area of CO and CH₄ detected by the TCD in the outlet gas.

The carbon balance was calculated as

$$C_{\text{balance}}(\%) = \frac{(A_{\text{CO}_2}^{\text{out}} \times f_{\text{CO}_2/\text{N}_2} + A_{\text{CO}}^{\text{out}} \times f_{\text{CO}/\text{N}_2} + A_{\text{CH}_4}^{\text{out}} \times f_{\text{CH}_4/\text{N}_2}) \times (1 - X_{\text{CO}_2})}{A_{\text{CO}_2}^{\text{out}} \times f_{\text{CO}_2/\text{N}_2}} \times 100\% \quad (3)$$

where $f_{\text{CO}_2/\text{N}_2}$ is relative correction factor of CO₂ to N₂, which is determined by the calibrating gas. $A_{\text{CO}_2}^{\text{out}}$ is the chromatographic peak area of CO₂ detected by the TCD in the outlet gas. The carbon balance was calculated to be greater than 98% in all the tests.

The relevant supplements have been added in the revised manuscript on page 20, line 19–27 and page 21, line 1–11 (highlighted in yellow). Thanks for the reviewer's valuable comments again.

Comment 5: *The units of apparent activation energy must be revised: KJ mol⁻¹ by kJ mol⁻¹.*

Response: Thanks for the reviewer's reminder. We have corrected KJ mol⁻¹ to kJ mol⁻¹. Thanks for the reviewer's valuable comments again.

To Reviewer 2:

Reviewer #2: The authors have submitted an interesting manuscript on a new catalyst for the RWGS reaction, based on platinum on Mo₂N that forms an MoO_x surface layer. This way an unusually high concentration of surface oxygen vacancies are formed that cooperate with the redox-active platinum sites.

The work is of high significance for the field because the presented catalyst excels the catalysts presented in literature so far by a much higher activity. Thermodynamic equilibrium conversion can be achieved at 300 °C already. The conclusions are sufficiently supported by the provided data to a large extent.

It can be expected that the insights provided by the authors could be helpful for the scientific community to better understand the principles of active and selective RWGS catalysts. The practical impact of the work cannot be assessed, since the long-term stability of the catalyst is unclear, the possibility to prepare a practically applicable catalyst remains open and the process economics for a catalyst working at low reaction temperatures with very limited equilibrium conversion to the desired products might be unfavorable. However, the provided scientific insights could justify publication in Nature Communications.

Comment 1: *page 3, top: "...efficient supported catalysts always require synergy between multiple components1–4." This is daring statement and should be better justified. "Synergy" means that all components profit from each other. However, in many cases of supported catalysts only the catalytically active component profits from the support, which cannot be called "synergy".*

Response: Thanks for the reviewer's valuable comments. We have modified the statement "efficient supported catalysts always require synergy between multiple components" to "high-performance supported catalysts always require the cooperation of multiple components". **The relevant supplements have been modified in the**

revised manuscript on *page 3, line 4–5 (highlighted in yellow)*. Thanks for the reviewer's valuable comments again.

Comment 2: page 3, middle: "the flexibility of surface oxygen atoms" What do the authors mean by "flexibility"? This term could mean almost everything and needs specification.

Response: Thanks for the reviewer's valuable comments. The statement "the flexibility of surface oxygen atoms on these oxides is very limited^{18–20}, which undoubtedly limits the creation of active metal-oxygen vacancy synergistic sites." has been changed to "due to the stability of the surface oxygen atoms, the ability to generate surface oxygen vacancies in these oxides have been restricted^{18–20}, which undoubtedly limits the creation of active metal-oxygen vacancy synergistic sites." **The relevant supplements have been modified in the revised manuscript on page 3, line 23–25 (highlighted in yellow)**. Thanks for the reviewer's valuable comments again.

Comment 3: page 3, bottom: "due to their unique electronic structure"⁹. Please specify what is "unique".

Response: Thanks for the reviewer's valuable comments. As a typical transition-metal nitride, Mo₂N was formed by the interstitial incorporation of N in the Mo metal lattice. The interstitial incorporation of N resulted in an increase in the atomic spacing of metal Mo and the contraction of the *d*-band, making Mo₂N exhibit the noble-metal-like electronic structure. **The relevant supplements have been added in the revised manuscript on page 3, line 26–29 (highlighted in yellow)**. Thanks for the reviewer's valuable comments again.

Comment 4: page 3, bottom: The term "deoxidize" is used several times in the manuscript. This term is unknown to me. I would call this "reduce".

Response: Thanks for the reviewer's valuable comments. According to reviewer's suggestion, the term "deoxidize" has been revised to "reduce" in the revised version. Thanks for the reviewer's valuable comments again.

Comment 5: page 4, top: "However, CO₂ molecule is very stable, and the RWGS is endothermic, making the conversion of CO₂ at low temperature very challenging." I wonder how attractive low-temperature RWGS is, since the equilibrium is shifted to the reactants at low temperatures, resulting in low maximum conversions. Maybe more stable RWGS catalysts at higher temperatures are more attractive and desirable from an application point of view.

Response: Thanks for the reviewer's valuable comments. The RWGS reaction process is an important intermediate step in the synthesis process from CO₂ to methanol, which usually takes place at relatively low temperature (*Nat. Catal.* **2021**, 4, 242–250). Therefore, the catalyst with high efficiency for the low-temperature RWGS reaction could probably benefit the conversion of CO₂ to methanol. Besides, because the equilibrium of the RWGS reaction was limited at low temperatures, previous reported catalysts were almost inactive at low temperatures. In this work, we are committed to exploiting the activity limit of catalysts in the low-temperature RWGS reaction. At the same time, as stated by the reviewer, we are simultaneously focusing on the development of efficient Mo₂N-based catalysts for catalyzing the high-temperature RWGS reaction.

Besides concentrating on the enhancement of catalytic performance for RWGS, we also proposed a new strategy for constructing high-density metal cluster-oxygen vacancy synergistic catalytic sites by utilizing the stress of heterostructure. The unmatched low-temperature activity of Pt-MoO_x/Mo₂N proved that the strategy of constructing high-density cluster-vacancy sites was effective for enhancing catalytic performance. We hope that the presented strategy in this work can also be used to guide the construction of other effective supported catalysts for other catalytic reactions. Thanks for the reviewer's valuable comments again.

Comment 6: page 4, top: "...for the RWGS catalysts, active metal clusters can dissociate H_2 , while the vacancies can activate CO_2 . Therefore, the creation of catalysts with high-density synergistic sites between active metal cluster and oxygen vacancy may greatly improve the low-temperature activity of catalysts." Usually, only one reaction step is the rate-determining step and the authors should comment which one determines the overall reaction rate. In this regard, the authors should explain what they mean with the vage term "synergistic sites".

Response: Thanks for the reviewer's valuable comments. The dissociation of H_2 on the Pt clusters and the activation of CO_2 at the oxygen vacancy were both critical steps in the entire reaction process. However, they did not directly determine the overall reaction rate. In other words, neither of the two was the rate-determining step. Our calculations showed that H_2 molecules spontaneously dissociated on the Pt clusters, and the occurrence of the CO_2 adsorption and activation merely needed to overcome a very low energy barrier (0.30 eV). If we took the carboxyl route in Figure R6 as an example, the rate-determining step was the one that corresponds to TS3, i.e., the formation of an H_2O molecule via the combination of a hydroxyl adsorbate and an adsorbed hydrogen atom. The energy barrier of this step was calculated to be 1.03 eV.

Figure R6. Energy profiles of the three reaction routes (redox, carboxyl, and formate) on Pt_4-MoO_x , depicted in gray, red, and cyan, respectively. The black line represents

the common parts of the three pathways. The configurations of intermediates and transition states (TS) in the carboxyl route (energetically the most favorable according to the simulations) are displayed at the bottom.

It was worth noting that the occurrence of this rate-determining step was closely related to the H₂ dissociation and the CO₂ activation. The generation of the hydroxyl group came from the cleavage of the C-O bond ($\text{COOH}^* \rightarrow \text{CO}^* + \text{OH}^*$), while the CO₂ activation (reflected in the bending of the molecular configuration and the acquisition of the electric charge) and the bonding of CO₂ with hydrogen atoms (generated via H₂ dissociation) both promoted this process. Besides, the hydrogen atom that combined with the hydroxyl group also came from the dissociation of H₂. Thus, one could see that without the Pt clusters working together with the oxygen vacancies, the occurrence of the RWGS reaction would be, if not impossible, very difficult. This was also what we meant by using the word “synergistic”.

Comment 7: Page 5, middle: "isolated Pt atoms were anchored on the γ -Mo₂N support (Figure 1b and Supplementary Figure 2)." The isolated Pt atoms cannot be seen in the presented figures, although they are marked. Based on the shown figures, the existence of the isolated platinum atoms depend on belief. Scientific evidence has to be provided.

Response: Thanks for the reviewer’s valuable comments. HAADF-STEM is the Z contrast image, which is related to the atomic number. And the brighter part in HAADF-STEM images indicates the atom with the bigger atomic number. Since the atomic number of Pt is much larger than that of Mo, it is likely that these bright spots are Pt atoms (Figure R7) (*Nature* **2021**, 589, 396–401). Meanwhile, the EXAFS spectra of Pt L₃ edge over fresh 0.5Pt-MoO_x/Mo₂N only showed Pt-O contribution, suggesting the Pt species existed in the form of isolated Pt atoms. Therefore, it could be inferred that the bright spots (noted by red cycles) in Figure R8 were Pt atoms. The corresponding images have been modified in the revised manuscript and the revised supporting information. Thanks for the reviewer’s valuable comments again.

Figure R7. High-resolution STEM Z-contrast images of fresh 2 wt% (Pt₁-Pt_n)/α-MoC catalyst, with some of the Pt₁ species highlighted by the red dashed circles, Pt clusters (Pt_n) highlighted by the green ellipses. Scale bars, 1 nm. Insets: schematic representations of the Pt₁-Pt_n/α-MoC catalyst (Pt, yellow; Mo, blue; C, black). (from *Nature* **2021**, 589, 396–401).

a

Figure R8. The aberration-corrected HAADF-STEM images of the fresh 0.5Pt-MoO₃/Mo₂N catalyst.

Comment 8: page 6, middle: "the single Pt atoms in 0.5Pt-MoO₃/γ-Mo₂N agglomerated into nanoclusters during the RWGS reaction" See my comment above. The agglomeration is also invisible from the presented figures. Scientific evidence has to be provided.

Response: Thanks for the reviewer's valuable comments. As shown in Figure R9, based on the contrast difference between Pt and Mo, the bright area in aberration-corrected HAADF-STEM images of the used 0.5Pt-MoO₃/Mo₂N catalyst could be assigned to Pt clusters, which consisted of a few of Pt atoms. And the Pt-Pt coordination observed in the EXAFS spectra of the used 0.5Pt-MoO₃/Mo₂N catalyst also confirmed the presence of Pt clusters. The corresponding images have been modified in the revised supporting information. Thanks for the reviewer's valuable comments again.

a

C

Figure R9. The aberration-corrected HAADF-STEM images of the used 0.5Pt-MoO₃/Mo₂N catalyst.

Comment 9: page 8, middle: "0.5Pt-MoO_x/γ-Mo₂N showed excellent stability after 300 h reaction". The authors should be more honest in the description of the measured data. I agree that the catalyst is relatively stable, but I still see a slight deactivation trend over 300 h.

Response: Thanks for the reviewer's valuable comments. The statement "0.5Pt-MoO_x/γ-Mo₂N showed excellent stability after 300 h reaction" has been changed to "0.5Pt-MoO_x/Mo₂N showed good stability, and it could maintain ~80% initial CO₂ conversion after 300 h reaction at 300 °C with a space velocity of 300,000 ml·g_{cat}⁻¹·h⁻¹". **The relevant supplements have been modified in the revised manuscript on page 8, line 11–13 (highlighted in yellow).** Thanks for the reviewer's comments again.

Comment 10: page 11, middle: "After the sample was treated by NH₃ flow at 300 °C, the proportion of oxygen content decreased, suggesting that the NH₃ flow converted part of the oxide into Mo₂N (Figure 3d and Supplementary Figure 19–21)" It is unclear for the reader how to see this claim from Figure 3d. A more detailed argumentation is required.

Response: Thanks for the reviewer's valuable comments. A more detailed argumentation is given as follows. As shown in Figure 3d, after the treatment by NH₃, the Raman peaks of MoO_x existed in the RWGS reaction process disappeared, which suggested the nitridation of the MoO_x structure. After that, under the effect of H₂O, the MoO_x structure was regenerated. However, as shown in Supplementary Figure 19, the broken MoO_x structure could not be regenerated in the RWGS reaction, indicating that the H₂O generated in the RWGS reaction could leave the catalyst surface quickly without oxidizing the catalyst surface. Besides, the color change of the MoO₃ sample treated with NH₃ flow and the quasi *in situ* XPS spectra of 0.5Pt-MoO_x/Mo₂N (Supplementary Figure 20 and 21) further confirmed that the NH₃ flow converted a part of the oxide into Mo₂N. **The relevant supplements have been added in the revised manuscript on page 11, line 19–28 (highlighted in yellow).** Thanks for the reviewer's valuable comments again.

Comment 11: page 13, bottom: "CO gaseous signal" DRIFTS provide signals of adsorbed CO. If gaseous CO was indeed measured more information should be provided. How were the DRIFTS experiments designed? Was the gas phase over the catalyst measured as well?

Response: Thanks for the reviewer's valuable comments. **Due to the black color of the catalyst (Figure R10), we could not obtain any information of adsorbed species and reactant intermediate in the *in situ* DRIFTS spectra, except for the gaseous signals of CO and CO₂ (Figure R11).** The signals at 2174 cm⁻¹ and 2112 cm⁻¹ could be assigned to gaseous CO (*Nat. Catal.* **2021**, 4, 418–424; *Angew. Chem. Int. Ed.* **2016**, 55, 10606–10611). And the peaks at 2361 cm⁻¹ and 2341 cm⁻¹ could be attributed to

gaseous CO₂ (*J. Am. Chem. Soc.* **2019**, *141*, 4613–4623; *ACS Catal.* **2018**, *8*, 7455–7467).

Figure R10. The photograph of 0.5Pt-MoO₃/Mo₂N.

Figure R11. *In situ* diffused reflectance infrared Fourier transform spectroscopy (DRIFTS) spectra of 0.5Pt-MoO_x/Mo₂N during (a) CO₂ treatment and (b) reaction conditions at 300 °C, respectively.

The specific test steps of *DRIFTS* experiments are given as follows: All of the *in situ* DRIFTS spectra were collected by using a Bruker Vertex 70 FTIR spectrometer with a mercury cadmium telluride (MCT) detector cooled with liquid nitrogen. **And all tests were measured at atmosphere pressure.** The treatment process of CO₂ on 0.5Pt-MoO_x-Mo₂N was investigated by *in situ* DRIFTS measurement at 300 °C. Prior to the *in situ* DRIFTS measurement, ~30 mg sample was pretreated at 300 °C for 30 min under 5% H₂/Ar mixed gas. The background spectra were collected under N₂ atmosphere at 4 cm⁻¹ resolution at 300 °C. After the collection of the background spectrum, the mixed gas consisted of 2% CO₂/Ar was introduced into the chamber. Continuous recording of the IR profiles was maintained for 5 min. As for the test under RWGS reaction

conditions, after background acquisition, the reaction gas with 15% CO₂/30% H₂/55% N₂ was introduced into the *in situ* chamber. All DRIFTS results were analyzed by using OPUS software. **The gas phase over the catalyst was also measured.** Thanks for the reviewer's valuable comments again.

Comment 12: page 14, bottom: "We noted in passing that in experiments, which of the two routes would be taken in the reaction process was not identified." This sentence is hardly understandable. Please rephrase.

Response: Thanks for the reviewer's valuable comments. In the revised version, we have rephrased the sentence (as well as the one that follows) to "We noted in passing that from our experiments, we could not determine whether the actual reaction process was via the carboxyl route or the formate pathway. (But no matter which of the two was adopted, the cooperation of Pt clusters and oxygen vacancy would always play a crucial role in promoting the RWGS reaction.)" **The relevant contents have been modified in the revised manuscript on page 14, line 27–29 (highlighted in yellow).** Thanks for the reviewer's comments again.

Comment 13: page 16, middle: "concentrated hydrochloric acid". Please specify the concentration.

Response: Thanks for the reviewer's valuable comments. The mass fraction of the concentrated hydrochloric acid was 37 wt.%. **The relevant supplement has been added in the revised manuscript on page 16, line 27–28 (highlighted in yellow).** Thanks for the reviewer's comments again.

Comment 14: page 16, bottom: "ethyl alcohol". Please write ethanol instead.

Response: Thanks for the reviewer's valuable comments. The relevant content has been changed as required. Thanks for the reviewer's comments again.

Comment 15: page 17, top: "designed amount of chloroplatinic acid". This is unclear and should be rephrased. Please be specific with respect to the amounts.

Response: Thanks for the reviewer's valuable comments. In the preparation of Pt-MoO₃/Mo₂N catalysts, the amount of the γ -Mo₂N support was 400 mg, and the amount of chloroplatinic acid (0.19 mol/L) was 54 μ L. **The relevant supplement has been added in the revised manuscript on page 17, line 6 and 7 (highlighted in yellow).** Thanks for the reviewer's comments again.

Comment 16: page 17, middle: "half an hour" = 0.5 h.

Response: Thanks for the reviewer's valuable comments. The relevant content has been revised. Thanks for the reviewer's comments again.

Comment 17: page 17, middle: "ultrasonic". Should read "ultrasonicated".

Response: Thanks for the reviewer's valuable comments. The "ultrasonic" has been revised to "ultrasonicated". Thanks for the reviewer's comments again.

Comment 18: page 17, middle: "HCL aqueous solution". Please specify concentration.

Response: The reviewer's comment is highly appreciated by us. The mass fraction of HCl in the HCl aqueous solution was 6.2%. **The relevant supplement has been added in the revised manuscript on page 17, line 10 (highlighted in yellow).** Thanks for the reviewer's comments again.

Comment 19: page 17, middle: "according to previous reports". Please add reference.

Response: Thanks for the reviewer's valuable comments. The relevant reference has been added. Thanks for the reviewer's comments again.

Comment 20: middle: "certain amount of chloroplatinic acid solution". Please specify amount and concentration.

Response: Thanks for the reviewer's valuable comments. The amount of used chloroplatinic acid (0.19 mol/L) was 68 μ L. **The relevant supplement has been added in the revised manuscript on page 17, line 19 (highlighted in yellow).** Thanks for the reviewer's comments again.

Comment 21: page 17, middle: "sodium carbonate solution". Please specify concentration.

Response: Thanks for the reviewer's valuable comments. The concentration of sodium carbonate solution was 0.1 mol/L. **The relevant supplement has been added in the revised manuscript on page 17, line 21 (highlighted in yellow).** Thanks for the reviewer's comments again.

Comment 22: page 19, top: "clear away". Please rephrase.

Response: Thanks for the reviewer's valuable comments. "Clear away" has been changed to "purge". Thanks for the reviewer's comments again.

Comment 23: page 19, top: "physical adsorbed CO₂ molecules" should ready "physically adsorbed CO₂ molecules".

Response: The reviewer's comment was highly appreciated by us. The relevant content has been revised as required. Thanks for the reviewer's comments again.

Comment 24: Supplementary Figure 9: Please clearly label the figures, e.g. what is fresh and used? What does e and f show? Moreover, the spectra are only superficially explained in the main manuscript.

Response: Thanks for the reviewer's valuable comments. The relevant contents have been revised as required. Fresh and used catalysts meant the catalysts before and after the RWGS reaction. Figures e and f presented the wavelet transformation (WT) EXAFS oscillation of Pt L3 edge in 2Pt-MoO₃/Mo₂N catalysts before and after the RWGS reaction. Besides, the relevant description of Supplementary Figure 9 has been supplied

as follows. Wavelet transformation (WT) EXAFS oscillations of Pt L3 edge exhibited an increase in the intensity of Pt-Pt peak of the catalysts after the RWGS reaction compared to the catalysts before the reaction, which further indicated the aggregation of Pt atoms. **The relevant contents have been modified in the revised supporting information on page S13 (highlighted in yellow).** Thanks for the reviewer's comments again.

Comment 25: page S14: "For all Pt-MoO₃/Mo₂N catalysts with different Pt loading, the stronger intensity of the Pt-Pt coordination peak of the fresh catalysts than that of the used catalysts was confirmed, suggesting the reduction and aggregation of the Pt species in the RWGS reaction." From the figure, I see that the Pt-Pt coordination peak of the used catalyst is stronger, which is in conflict with the authors statement.

Response: Thanks for the reviewer's valuable comments. **We have revised the relevant statement in the revised supporting information on page S14, line 6 (highlighted in yellow).** Thanks for the reviewer again.

Comment 26: Supplementary Figure 14. Labels a) and b) are mixed.

Response: Thanks for the reviewer's valuable comments. According to reviewer's comment, the relevant labels have been corrected. Thanks for the reviewer's comment again.

Comment 27: S22, bottom: "... that the surface of MoO₃ is difficult to deoxygenate during the RWGS reaction and generate a lot of oxygen vacancies." The listed consequences are in obvious conflict with each other. Please rephrase.

Response: Thanks for the reviewer's valuable comments. The relevant statement has been changed to "When the MoO₃ sample was treated with RWGS reaction gas, the signal of MoO₃ did not disappear and no signal of MoO_x was generated, which indicated that the surface of MoO₃ was difficult to be reduced during the RWGS reaction, and thus no oxygen vacancy could be generated." **The relevant contents have been**

modified in the revised supporting information on page S22, line 4–7 (highlighted in yellow). Thanks for the reviewer's comment again.

Comment 28: Supplementary Figure 22. What does "Bader charge" mean?

Response: Thanks for the reviewer's valuable comments. In the original version, the "Bader charge" was meant to the calculated number of charges carried by each atom, which was obtained via the Bader charge analysis. This was a method named after Richard Bader, by which one could assign the calculated total charge density of the whole system to each atom. Thus, **by calculating the difference between the number of electrons assigned and the number of the corresponding valence electrons, one could obtain the number of charges carried by each atom.**

In the revised version, we have changed the expression "q: Bader charge of CO₂" to "q: calculated number of charges carried by CO₂ via the Bader charge analysis". Thanks for the reviewer's comment again.

Comment 29: Supplementary Figure 23. A catalyst can be pretreated but not an activity test.

Response: Thanks for the reviewer's valuable comments. The figure note has been changed to "The catalytic performance of 0.5Pt-MoO_x/Mo₂N after the pretreatment with 5% H₂/Ar and 1% O₂/Ar, respectively." **The relevant contents have been modified in the revised supporting information on page S27, line 2 (highlighted in yellow).** Thanks for the reviewer's comment again.

Comment 30: Supplementary Figure 24: This figure is unclear. What was measured under what conditions? Which intensities are given at the y-axis.

Response: Thanks for the reviewer's valuable comments.

(1) The CO₂ dissociation experiment (Figure R12a) was conducted to determine whether CO₂ could be dissociated directly into CO without the assistance of H₂. The specific test steps were as follows.

The 0.5Pt-MoO_x/Mo₂N catalyst was firstly pretreated by 5% H₂/Ar (30 mL/min) at 300 °C for 30 min, and then flushed with Ar gas flow (30 mL/min) at room temperature for 30 min. After that, the CO₂ dissociation experiment was carried out under 2% CO₂/Ar (30 mL/min) from room temperature to 300 °C. The mass-to-charge ratio signals of CO₂ (m/z=44) and CO (m/z=28) were collected online.

As shown in Figure 12a, during the whole test process, no CO signal was detected, suggesting that CO₂ could not be dissociated directly to CO.

(2) The temperature-programmed surface reaction (TPSR) (Figure R12b) was measured to detect the dissociation of CO₂ with the assistance of H₂. The specific test steps were as follows.

Firstly, the 0.5Pt-MoO_x/Mo₂N catalyst was treated with 5% H₂/Ar (30 mL/min) at 300 °C for 30 min. Then, use Ar gas flow (30 mL/min) to purge the sample at room temperature for 30 min. After that, the TPSR measurement was carried out under the RWGS reaction atmosphere (23% CO₂, 69% H₂, 8% N₂) (30 mL/min) from room temperature to 300 °C. The mass-to-charge ratio signals of CO₂ (m/z=44) and CO (m/z=28) were collected online.

As illustrated in Figure R12b, with the temperature increased, the CO₂ signal gradually decreased, while the CO signal gradually increased. This result indicated that H₂ was involved in the conversion process of CO₂ to CO.

(3) The *in situ* Diffuse Reflectance Infrared Fourier Transform Spectroscopy (DRIFTS) was measured to explore whether the dissociated H was directly involved in the conversion process from CO₂ to CO (Figure R12c and d). The absorbance of infrared light with continuous wavelength was collected online. The testing process was as follows. Prior to the *in situ* DRIFTS measurement, ~30 mg sample was pretreated at 300 °C for 30 min under 5% H₂/Ar mixed gas. After that, the 5% H₂/Ar was switched to pure N₂. The background spectra were collected under N₂ atmosphere at 4 cm⁻¹ resolution at 300 °C. After the collection of the background spectrum, the mixed gas consisted of 2% CO₂/Ar and was introduced into the chamber. Continuous recording of the IR profiles was maintained for 5 min. As for the test under RWGS reaction

conditions, after background acquisition, the reaction gas with 15% CO₂/30% H₂/55% N₂ is introduced into the *in situ* chamber.

As shown in Figure R12c, CO₂ could not be dissociated directly into gaseous CO. However, the signal of gaseous CO was detected with the assistance of H₂ (Figure R12d), which again indicated that H₂ was involved in the conversion process of CO₂ to CO. Above results suggested that the RWGS reaction catalyzed by 0.5Pt-MoO_x/Mo₂N might follow the associative mechanism rather than the redox mechanism. **The relevant figures have been modified in the revised supporting information on page S28 (highlighted in yellow). Thanks for the reviewer's valuable comments again.**

Figure R12. (a) The CO₂ dissociation experiment of 0.5Pt-MoO_x/Mo₂N. (b) Temperature programmed surface reaction (TPSR) result of 0.5Pt-MoO_x/Mo₂N. (c, d) *In situ* diffused reflectance infrared Fourier transform spectroscopy (DRIFTS) spectra of 0.5Pt-MoO_x/Mo₂N during CO₂ treatment and reaction conditions at 300 °C, respectively.

Comment 31: Supplementary Figures 25 and 26. There is not reference to these figures in the manuscript.

Response: Thanks for the reviewer's valuable comments. According to reviewer's comment, we have revised the corresponding content. **The relevant contents have been added in the revised manuscript on page 14, line 9 and 27 (highlighted in yellow).** Thanks for the reviewer again.

REVIEWERS' COMMENTS

Reviewer #1 (Remarks to the Author):

The authors have addressed correctly the comments. The article is now suitable to be published.

Reviewer #2 (Remarks to the Author):

The authors have answered all questions convincingly and the manuscript can be recommended for publication in the journal Nature Communication.

Responses to the Reviewers' Comments and the Corresponding Revisions

Reviewer #1:

Comment: The authors have addressed correctly the comments. The article is now suitable to be published.

Response: Thanks for reviewer's valuable comments. According to reviewer's comments, we have carefully revised and supplemented the manuscript, which undoubtedly greatly improved the quality of our research work. Thanks for the reviewer again.

Reviewer #2:

Comment: The authors have answered all questions convincingly and the manuscript can be recommended for publication in the journal Nature Communication.

Response: The professional and detailed comments and suggestions of the reviewers are important for the improvement of our research work. According to the reviewer's comments, the quality of our manuscript has been greatly improved compared to the original version. Thanks for the reviewer again.